# Epigenetic regulation of neural stem cell aging in the mouse hippocampus by Setd8 downregulation

Shuzo Matsubara[1,7], Kanae Matsuda-Ito[1,2], Haruka Sekiryu[1], Hiroyoshi Doi ![ORCID][1], Takumi Nakagawa[1], Naoya Murao ![ORCID][3], Hisanobu Oda[4], Kinichi Nakashima ![ORCID][1✉] & Taito Matsuda ![ORCID][1,2,5,6,7✉]

## Abstract

**Neural stem cells (NSCs) in the mammalian brain decline rapidly with age, leading to impairment of hippocampal memory function in later life. However, the relationship between epigenetic remodeling and transcriptional regulation that compromises hippocampal NSC activity during the early stage of chronological aging remains unclear. Here, we performed single-cell RNA sequencing (scRNA-seq) and single-cell ATAC sequencing (scATAC-seq) on NSCs and newly generated neurons across different stages. Integrated data analysis revealed continuous alterations in the chromatin profile of hippocampal NSCs and their progeny from neonatal to mature adult stages, accompanied by consistent changes in transcriptional profiles. Further, decreased expression of *Setd8*, encoding the enzyme for histone H4 monomethylation at lysine 20 (H4K20me1), underlies age-related changes in mouse hippocampal NSCs. Notably, depletion of *Setd8* elicits alterations in gene expression and epigenetic regulation that phenocopy age-related changes, and impairs NSC activity, leading to hippocampal memory deficits. Together, our study provides a global map of longitudinal chromatin and transcriptome changes during brain aging and identifies mechanistic insights into early-onset decline of NSC activity and hippocampal neurogenesis that precedes functional aging.**

**Keywords** Neural Stem Cell; Aging; Epigenetics; Hippocampus
**Subject Categories** Chromatin, Transcription & Genomics; Molecular Biology of Disease; Neuroscience

## Introduction

Aging is the slow but gradual decline in the function of cells and organs due to the accumulation of cellular damage. This process has been studied historically by comparing organisms in mature adult and middle age with those in old age. More recent studies have highlighted approaches to prevent aging at a younger age to promote tissue function later in life by identifying mechanisms that initially compromise cellular behaviors (Zhao et al, 2008; Aimone et al, 2014; Gillman, 2005; Belsky et al, 2015).

Loss of epigenetic but not genetic information as a potential cause of aging emerged from yeast studies (Kennedy et al, 1997; Sinclair and Guarente, 1997). Epigenetic changes linked to aging, including changes in DNA methylation and histone modifications, such as H3K4me3, H3K9me3, and H3K27me3, have now been observed in multicellular organisms, including mice and humans (Benayoun et al, 2015; Sen et al, 2016; Pal and Tyler, 2016). In addition to the link between epigenetics and aging, a recent study has demonstrated that loss of epigenetic information causes mammalian aging (Yang et al, 2023). Epigenetic modifications are often reversible with the aid of epigenetic regulators, which lays the theoretical basis for aging regulation and may, therefore, be a promising target for aging intervention strategies.

Brain function declines with age. In parallel, incidence rates for neurodegenerative diseases, such as Alzheimer's disease and Parkinson's disease, and brain tumors, such as gliomas, rapidly increase in the elderly population. While aging affects all cell types in the brain and can cause physiological decline and disease, neural stem cells (NSCs) in the adult brain have the potential to generate new neurons (neurogenesis) and regenerate brain function. Quiescent NSCs expressing Nestin, Gfap, and Sox2 divide slowly in the subgranular zone (SGZ) of the hippocampal dentate gyrus (DG) and serve as a source for neurons (Gonçalves et al, 2016). Once quiescent NSCs are activated, they produce neural progenitor cells (NPCs), which subsequently differentiate into immature neurons (IMNs). These IMNs eventually become mature granule neurons responsible for processing hippocampus-dependent memory (Zhao et al, 2008; Aimone et al, 2014). However, NSC numbers are ever-changing throughout life. Although a small fraction persists for a lifetime, NSCs decline rapidly in the hippocampus of mice during the period within 24 weeks after birth, and the decline then becomes very slow; these changes result in poor hippocampal memory function and may underlie aging-related human disorders such as Alzheimer's disease in later life (Moreno-Jiménez et al, 2019). In addition to a decrease in the number of NSCs, aging impairs NSC proliferation, neuronal differentiation, and newborn neuron survival (Ben Abdallah et al, 2010; Bondolfi

[1]Stem Cell Biology and Medicine, Department of Stem Cell Biology and Medicine, Graduate School of Medical Sciences, Kyushu University, Fukuoka 812-8582, Japan. [2]Laboratory of Neural Regeneration and Brain Repair, Division of Biological Science, Graduate School of Science and Technology, Nara Institute of Science and Technology (NAIST), Nara 630-0192, Japan. [3]Laboratory of Biochemistry and Molecular Biology, Department of Medical Sciences, University of Miyazaki, Miyazaki 889-1601, Japan. [4]Division of Integrative Medical Oncology, Saiseikai Kumamoto Hospital, Kumamoto 861-4193, Japan. [5]Life Science Collaboration Center (LiSCo), NAIST, Nara 630-0192, Japan. [6]Data Science Center, NAIST, Nara, Japan. [7]These authors contributed equally: Shuzo Matsubara, Taito Matsuda. ✉E-mail: nakashima.kinichi.718@m.kyushu-u.ac.jp; matsuda.taito@naist.ac.jp

et al, 2004). NSCs also exhibit molecular aging features, such as altered proteostasis and inflammation (Navarro Negredo et al, 2020). Recent studies have shown that extracellular factors derived from niche resident cells, such as cytokines (Mosher and Wyss-Coray, 2014), as well as intracellular factors including nuclear membrane proteins, transcription factors, and epigenetic factors in NSCs (bin Imtiaz et al, 2021; Bedrosian et al, 2021; Ibrayeva et al, 2021; Kobayashi et al, 2019; Leeman et al, 2018; Schäffner et al, 2018; Vonk et al, 2020; Li et al, 2024), alter in aged animals when compared to their younger counterparts. Among epigenetic factors, for instance, downregulation of Tet2 in the hippocampus has been shown to impair the epigenetic process of DNA demethylation, resulting in decreased neurogenesis in aged mice (Gontier et al, 2018). In addition, the great decline of neurogenesis in the hippocampus initiates at an early stage in the mature brain of rodents and also human (Moreno-Jiménez et al, 2019; Ben Abdallah et al, 2010; Bondolfi et al, 2004), suggesting that NSCs can age molecularly at the early stages of chronological aging with epigenetic changes.

However, it remains largely unclear when and how transcriptional changes are linked to epigenetic remodeling in NSCs during the early stages of chronological aging. Through integrated analysis of single-cell RNA sequencing (scRNA-seq) and single-cell assay for transposase-accessible chromatin sequencing (scATAC-seq), we demonstrate in this study that the chromatin landscape undergoes continuous alterations from the early stages of chronological aging in hippocampal NSCs, resulting in corresponding transcriptional profile changes. The monomethylation of histone H4 at lysine 20 (H4K20me1), which is known to be associated with cellular proliferation and senescence, is mediated exclusively by a single enzyme, Setd8 (Nishioka et al, 2002; Tanaka et al, 2017; Wickramasekara and Stessman, 2019; Oda et al, 2009; Beck et al, 2012; Huang et al, 2021). We found that the expression of Setd8 decreased with age in hippocampal NSCs. This reduction aligns with the observed decrease in H4K20me1 histone modification level. The specific deletion of Setd8 in NSCs led to a depletion of NSCs, thereby decreasing neurogenesis and impairing hippocampal memory function. Lentivirus-mediated Setd8 downregulation recapitulated age-related transformations in the transcriptomic and epigenomic profiles of NSCs. We thus elucidate an epigenetic mechanism that mediates NSC aging in the mammalian hippocampus during early chronological aging.

## Results

### scRNA-seq reveals age-dependent molecular signatures of NSCs and new neurons

To investigate transcriptomic alterations in NSCs and their progeny during early stages of chronological aging, we employed Nestin-EGFP mice (Yamaguchi et al, 2000) and isolated Nestin-EGFP+ cells from the dentate gyrus (DG) of their hippocampus at neonate (postnatal day 5: P5), adult (12-week-old: 12w), and mature adult (24-week-old: 24w) stages using flow cytometry and performed scRNA-seq with the 10X Genomics platform (Appendix Fig. S1A; Fig. 1A). We obtained transcriptome datasets of 15,232 cells and 18,308 genes across cells passing quality control and detected 17 clusters (Appendix Fig. S1B–K). Consistent with a previous report

(Shin et al, 2015), Nestin-EGFP+ cells included cells other than NSCs and their progeny, such as vascular cells (Appendix Fig. S1I–K). We then re-clustered cells only in astrocyte or astrocyte precursor cell (AS), NSC, neural progenitor cell (NPC), and immature neuron (IMN1 and IMN2) clusters (Fig. 1B,C). NSCs were identified by their close similarity to astrocytes (Gfap and Sox2) and the higher expression of the NSC marker Hopx (Appendix Fig. S1D,L). They almost lacked expression of Aqp4 and S100b (Appendix Fig. S1C,L), markers for astrocytes. NPCs generated from NSCs are a proliferative progenitor population and express a strong cell proliferation-associated gene, Hmgb2 (Kimura et al, 2018) (Appendix Fig. S1E,L). Both IMN1 and IMN2 clusters exhibit the expression of Tubb3 and Dcx (Appendix Fig. S1L). The IMN1 cluster retains Eomes expression, which is shared with NPCs, while the IMN2 cluster shows higher expression of Calb2 compared to the IMN1 cluster (Appendix Fig. S1F,L). Marker gene expression patterns in each cluster are thus similar to those found in previous reports (Hochgerner et al, 2018; Ximerakis et al, 2019; Shin et al, 2015).

Next, we identified cluster-specific marker genes in the cells during adult neurogenesis (from NSC through IMN1 to IMN2) and found that the expression of these genes changes only slightly with age, at least by 24w (Fig. 1C,D; Dataset EV1), suggesting that NSCs and their progeny preserve their inherent cellular identities as they age. To explore why NSC activity and neurogenesis are reduced with age, we identified genes whose changed expression in NSCs and their progeny is associated with age progression. In the NSC cluster, there were 853 differentially expressed genes (DEGs), of which 568 and 285 were commonly up- and downregulated, respectively, at 12w and 24w compared to P5 (hereafter referred to as up- and downregulated DEGs) (Fig. 1E; Dataset EV2), indicating that aging alters gene expression in NSCs with little loss of their cellular identity. These up- and downregulated DEGs in NSCs tended to gradually increase and decrease in expression with age, respectively (Appendix Fig. S1M,N). These data suggest that these changes in gene expression progress gradually from early postnatal life. When we subjected these genes to biological process analyses through gene ontology (GO), genes whose expression increased with age were associated with metabolic processes (Fig. 1I), consistent with a previous report (Ibrayeva et al, 2021). On the other hand, decreased genes were associated with nervous system development, suggesting that these changes in gene expression affect NSC activity. In support of this, Lmnb1, whose decreased expression is involved in the functional decline of NSCs in aged mice (bin Imtiaz et al, 2021; Bedrosian et al, 2021), was downregulated in NSCs with age (Appendix Fig. S1O). Apoe expression, which is a reported driver of cellular senescence (Kovtonyuk et al, 2019; Sousa-Victor et al, 2014), was increased with age (Appendix Fig. S1O). We further found that aging intensifies interferon signaling (Irf1, Ifit2, Ifit3, Ifitm1, and Ifitm3) in NSCs (Appendix Fig. S1P). Enhanced signaling of interferon, known as a member of the SASP family, characterizes cellular senescence (Hopfner and Hornung, 2020), suggesting that NSCs undergo some degree of cellular senescence by 24w. Previous studies have shown that aging promotes quiescence of NSCs in the DG (Ben Abdallah et al, 2010; Ibrayeva et al, 2021). We observed decreased expression of cell cycle regulators, including Cdk1 and Mcm5. The expression of quiescence-inducing genes, including Id4, Notch2, Huwe1, and Mfge8, was upregulated with age

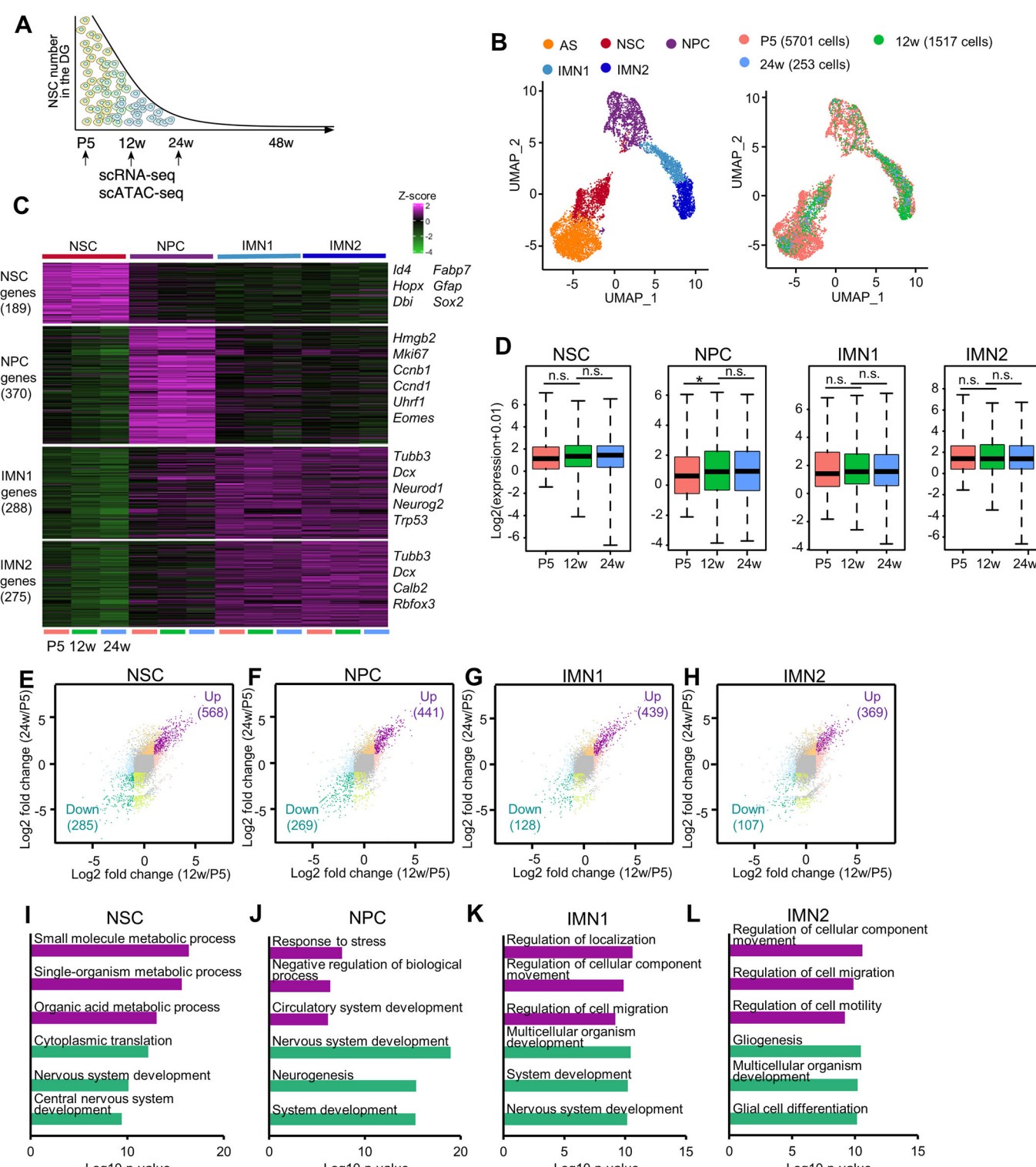

(Appendix Fig. S1Q), in agreement with previous studies (Zhou et al, 2018; Harris et al, 2021). Moreover, there was a significant overlap between the 568 upregulated or 285 downregulated DEGs in NSCs with age and genes related to quiescent or active NSCs, respectively (Appendix Fig. S1R,S), indicating the advancement of NSC quiescence with age. Thus, during the early stages of

chronological aging, NSCs display altered gene expression, which is linked to cellular quiescence and senescence.

Next, we asked whether gene expression in NPCs and neurons also changed with age, and identified DEGs in each cell type at 12w and 24w compared to P5 (Fig. 1F–H; Dataset EV2). GO analysis revealed that downregulated genes in NPCs are associated with

◀ **Figure 1. scRNA-seq analysis reveals continuous transcriptomic alterations in NSCs and progeny with age.**

(A) Schematic timeline for investigating gene expression and chromatin changes in hippocampal NSCs with age by scRNA-seq and scATAC-seq. (B) Uniform Manifold Approximation and Projection (UMAP) plot showing the re-clustered 7471 cells including astrocytes or astrocyte precursor cells (AS), neural stem cells (NSC), neural progenitor cells (NPC), immature neurons 1 (IMN1), and immature neurons 2 (IMN2). (C) Heatmap showing the average expression of marker genes for each cell type at P5, 12w, and 24w. The brighter the magenta color, the higher the expression level. (D) Box plot of marker gene expression for indicated cell types at P5, 12w, and 24w. $*p = 0.04162$ by Wilcoxon rank-sum test. Box plots show the median (centre line), the 25th and 75th percentiles (bounds of box), and the minimum and maximum values (whiskers). $n = 189$ (NSC), 370 (NPC), 288(IMN1), 275 (IMN2). Exact $p$ values from left to right: $p = 0.1696$; $p = 0.1201$; $p = 0.04162$; $p = 0.2363$; $p = 0.2511$; $p = 0.2782$; $p = 0.2286$; $p = 0.2371$. (E–H) Scatter plots showing fold change of gene expression in each cell cluster between P5 and 12w (x axis) and between P5 and 24w (y axis). Dots in any color other than gray are DEGs ($q$-value < 0.05 by Seurat package with MAST and fold change ≥ 2) that differed at 12w or 24w compared to P5. Genes with increased and decreased expression in common at 12w and 24w compared to P5 are shown in purple and bright green, and defined as upregulated DEGs and down-regulated DEGs, respectively. (I–L) Functional annotation of upregulated DEGs (purple) and down-regulated DEGs (green) in common at 12w and 24w compared to P5 in (E). The top three GO terms in each gene group are displayed.

neurogenesis-related terms (Fig. 1J), in agreement with impaired neurogenesis with age. Upregulated genes in IMN1 and IMN2 clusters were associated with cell migration, whereas down-regulated genes in these clusters were associated with development and differentiation (Fig. 1K,L). These results suggest that not only NSCs but also their progeny change gene expression with age, resulting in reduced neurogenesis.

Taking these observations together, age-related transcriptome changes may contribute to the reduced activity of NSCs and their progeny with little effect on the expression of each cluster-specific marker gene. Furthermore, although neurogenesis declines with age, the new neurons that are still generated do not have an identical gene expression profile between life stages, which could affect hippocampus-dependent memory formation.

## Single-cell ATAC-seq reveals altered chromatin accessibility in NSCs and new neurons with age

To gain deeper insight into how transcriptomic changes are regulated by epigenetic alteration with age in NSCs and their progeny, we analyzed chromatin accessibility in EGFP-positive cells isolated from hippocampal DGs of Nestin-EGFP mice by scATAC-seq, using the same experimental scheme as in Fig. 1 and the 10X Genomics platform. After quality control, we obtained 27,585 cells for further analysis, with 203,829 peaks mapped to the nuclear genome and a median of 10,850 fragments per cell. To assess the similarities between individual cells, we performed unsupervised analyses, including dimension reduction and clustering, using Signac, resulting in 17 clusters (Appendix Fig. S2A). Next, to infer cell cluster identities, we first calculated the gene activity scores by summing the fragments in the gene promoter and gene body. We then transferred cell-type labels by integrating the scATAC-seq dataset with the transcriptome dataset obtained from Fig. 1 using the Seurat package (Appendix Fig. S2A) (Shu et al, 2022) and then re-clustered to include only NSC, NPC, AS, and IMN clusters (Fig. 2A). We examined the chromatin accessibility near several known cell type-specific marker genes (Fig. 2B; Appendix Fig. S2B–G). As expected, each marker locus was accessible in the cells expressing the respective marker genes. We found that the mRNA expression of neuronal genes, e.g., *Tubb3* and *Dcx*, was high in IMN1 and IMN2 clusters, and that these gene loci were already accessible in the NPC cluster but not in the NSC cluster before gene expression (Fig. 2B). This result suggests that NSCs in the hippocampus do not yet have the chromatin potential to

differentiate into neurons, but acquire it later as they differentiate into NPCs before neuronal gene expression begins during neuronal differentiation.

We next assessed chromatin accessibility near the up- and downregulated DEGs in clusters of NSCs and their progeny with age and observed increased and decreased accessibility, respectively (Fig. 2C). This suggests that changes in chromatin accessibility in NSCs and their progeny with age concord with the corresponding gene expression changes. To identify regions with altered chromatin accessibility across the whole genome, rather than at individual gene loci, we detected the differentially accessible regions (DARs) in the identified clusters with age using MACS call and defined gained-open or -closed DARs ($p < 0.05$ and fold change ≥ 1.5) at both 12w and 24w compared to P5. In NSCs, 751 DARs were found to have gained-open status, while 7,488 had gained-closed status (Fig. 2D,E). These findings indicated that chromatin accessibility changes occur during the early stages of chronological aging in NSCs, with more regions decreasing in accessibility. As in the case of NSCs, there were more loci with reduced than with increased chromatin accessibility in NPCs, while in IMNs, the numbers of decreased and increased loci were comparable (Fig. 2D,E). To further characterize these DARs, we searched for consensus sequences within the identified gained-open and -closed DARs in each cell cluster. Motif discovery analysis of gained-opened DARs in NSCs indicated that the top three enriched motifs are for retinoic acid-related orphan receptors (Rora, Rorb, and Rorc), which regulate cell differentiation, circadian rhythm, and metabolism (Fig. 2F). On the other hand, among the top three motifs enriched in the gained-closed regions, Klf15 and Nrf1 have been suggested to be involved in the maintenance of NSCs with neurogenic activities (Fig. 2F) (Ohtsuka and Kageyama, 2019; Liu et al, 2021). We also found that motifs of multiple NSC activity-regulating transcription factors, such as Hes1, Zeb1, Ascl1, and Nfix, but not NSC maintenance transcription factor Sox2, were enriched in gained-closed regions (Fig. 2G). Furthermore, the gained-closed DARs of NPC and IMN1 clusters were found to contain the highest enrichment of Neurod1 and Neurog2 motifs, which are essential genes for neurogenesis (Fig. 2F). Thus, these results suggest that age-related functional changes in NSCs and their progeny are associated with early molecular aging, which is represented by altered chromatin accessibility of these cells in the regulatory factor-binding regions of stemness and neurogenesis with corresponding transcriptional changes.

 

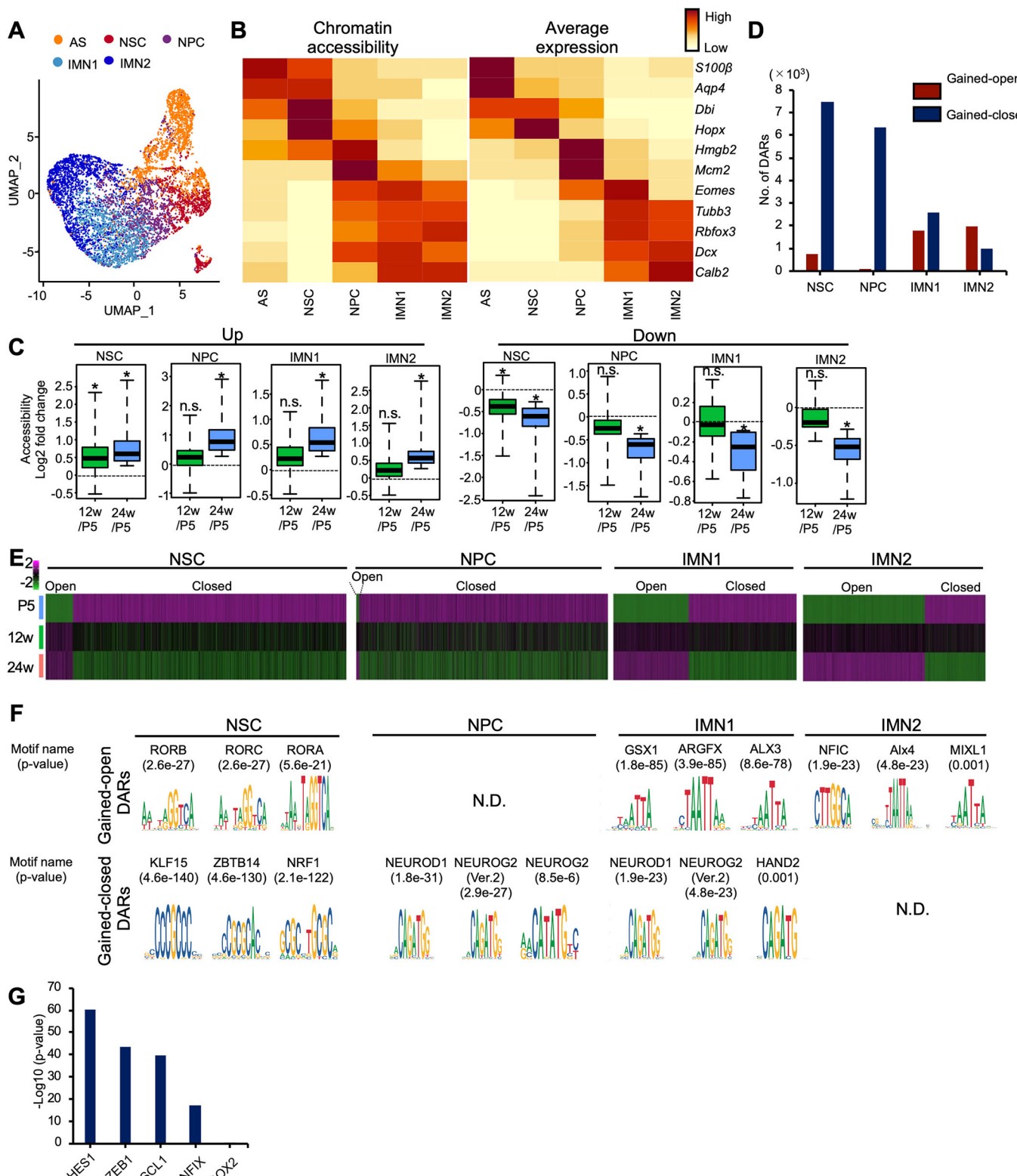

## Setd8 maintains the activity and neurogenic potential of NSCs

Having observed the transcriptomic alteration accompanied by chromatin remodeling with age, we explored the epigenetic regulators responsible for these changes. To this end, we mainly focused on NSCs, as age-related epigenomic and transcriptomic changes in NSCs may affect the behavior of their progeny and cause disturbances in subsequent neurogenic processes. By overlapping a previously defined list of epigenetics-related genes

**Figure 2. Altered chromatin accessibility during aging is associated with gene expression changes.**

(A) UMAP plot showing the re-clustered 7103 cells including AS, NSC, NPC, IMN1, and IMN2 identified by scATAC-seq. (B) Heatmap showing chromatin accessibility around the indicated genes (left) and corresponding gene expression levels (right) for each cluster. (C) Box plots showing the chromatin accessibility (gene activity score) around up- and downregulated DEG loci indicated in Fig. 1 (E–H) (≥1.25-fold, up in NSC cluster ($n = 244$), up in NPC cluster ($n = 184$), up in IMN1 cluster ($n = 159$), up in IMN2 cluster ($n = 129$), down in NSC cluster ($n = 184$), down in NPC cluster ($n = 84$), down in IMN1 cluster ($n = 19$), down in IMN2 cluster ($n = 26$)). To calculate the gene activity score, the gene coordinates were extracted and extended to include the upstream 2-kb region. *$p < 0.05$ by Wilcoxon rank-sum test compared with averaged gene expression at P5. n.s., not significant. Box plots show the median (centre line), the 25th and 75th percentiles (bounds of box), and the minimum and maximum values (whiskers). Exact $p$ values from left to right: $p = 0.0102$; $p = 0.01198$; $p = 0.145$; $p = 0.001434$; $p = 0.811$; $p = 0.0452$; $p = 0.8263$; $p = 0.0412$; $p = 0.03337$; $p = 0.0163$; $p = 0.1788$; $p = 0.03739$; $p = 0.2139$; $p = 0.0482$; $p = 0.2135$; $p = 0.03121$. (D) Bar graph showing chromatin accessibility in the number of gained-open or gained-closed DARs called by MACS ($p < 0.05$ by Wilcoxon rank-sum test and fold change ≥1.5). The number of accessible or inaccessible regions in common at 12w and 24w compared to P5 is shown. (E) Heatmaps showing the chromatin accessibility of the regions indicated in (D). (F) Enriched motifs for gained-open and -closed DARs. The top three enriched motifs in each group are displayed. (G) Bar graph showing the significant enrichment of motifs associated with NSC activity in the gained-closed DAR of NSCs with age.

(Matsuda et al, 2019) ($n = 171$) with the list of upregulated or downregulated DEGs in NSCs (Dataset EV3), we identified eight genes whose expression levels were significantly increased or decreased in 12w and 24w NSCs compared to P5 ($n = 8$, $q < 0.05$, fold change ≥ 1.5) (Fig. 3A; Appendix Fig. S3A,B). Of these eight genes, *Setd8*, the sole methyltransferase capable of mono-methylating histone H4 at lysine 20 (H4K20me1), showed the largest expression change from P5 to 24w of age (Appendix Fig. S3A,B). *Setd8* expression also tended to decrease in NPC, IMN1, and IMN2 clusters with age (Fig. 3B). To investigate the level of H4K20me1 modification in NSCs with age, we stained brain sections with antibodies for H4K20me1 and Nestin, and observed a decline in H4K20me1 levels in Nestin-positive (Nestin+) NSCs from P5 (Appendix S3C,D). To further analyze H4K20me1 levels in quiescent and active NSCs, we stained brain sections from Nestin-EGFP mice with antibodies for H4K20me1, Ki67, and EGFP, identifying EGFP+ NSCs with radial glia-like morphology in quiescent (Ki67−) and active (Ki67+) states (Fig. 3C,D). H4K20me1 levels decreased with age, consistent with *Setd8* expression, regardless of NSC activity state.

Setd8 has been reported to be associated with various genomic functions such as DNA replication, DNA repair, and cell division (Oda et al, 2009; Beck et al, 2012; Huang et al, 2021), although its role in the adult mouse hippocampus remains uncovered. We designed two shRNAs (KD_1 and KD_2) to determine whether Setd8 is crucial for NSC activity. Infection of cultured mouse NSCs with lentiviruses harboring these shRNAs decreased the expression of *Setd8* (Appendix Fig. S4A). *Setd8* knockdown (KD) reduced cultured NSC proliferation without affecting Caspase-dependent cell death (Appendix Fig. S4B,C). We also investigated the effects of *Setd8* KD using a lentiviral *Setd8* KD vector, as well as *Setd8* deletion using Cas9, on the proliferation of adult hippocampal NSCs. After infection, cells were subjected to puromycin selection to enrich successfully transduced cells. We found that both knockdown and knockout of *Setd8* reduced the proliferation of adult hippocampal NSCs (Appendix Fig. S4D–H). These results prompted us to focus on Setd8 as a prominent factor involved in the age-related functional decline of adult hippocampal NSCs in the following analysis.

To clarify the function of Setd8 in vivo, we specifically deleted *Setd8* in the NSCs of adult mice using the tamoxifen-inducible Cre recombinase (CreERT2) system (Imayoshi et al, 2006). Individuals of the *Setd8*^flox/flox^ mouse line (Oda et al, 2009) were crossed with *Nestin-CreERT2* mice (Imayoshi et al, 2006) and to a *Rosa26-YFP* reporter line. The resulting offspring were named *Setd8* conditional

knockout (cKO) and allowed fate mapping of NSCs that had undergone selective *Setd8* deletion. *Nestin-CreERT2* mice carrying the *Rosa26-YFP* reporter transgene but wild-type for *Setd8* were used as controls (Ctrl). We orally administered tamoxifen to 8-week-old *Setd8* cKO and Ctrl mice for four consecutive days and fixed the day after the last dose for analysis (Fig. 3E). H4K20me1 level was reduced in YFP+ NSCs of *Setd8* cKO mice compared to Ctrl mice in the adult hippocampus (Appendix Fig. S5A,B). We defined NSCs as cells harboring a radial Gfap+ process linked to a Sox2+ nucleus in the SGZ (Harris et al, 2021) and observed a significant reduction of the number and proportion of Ki67+ active NSCs in *Setd8* cKO mice compared with Ctrl mice (Fig. 3F,H,I). *Setd8* deficiency also decreased the number of Ki67− quiescent NSCs without promoting Caspase-dependent apoptotic cell death (Fig. 3G; Appendix Fig. S5C). Consistent with the reduction of NSCs, we found a decrease in the number of NPCs identified as Sox2+ and Ki67+ cells without a Gfap+ radial process in *Setd8* cKO mice (Fig. 3F,J). Next, to investigate whether *Setd8* depletion in adult NSCs impairs hippocampal neurogenesis, *Setd8* cKO and Ctrl mice were fixed six weeks after tamoxifen administration (Appendix Fig. S5D). This showed that the number of YFP+ and Dcx+ newly generated immature neurons was markedly reduced in the *Setd8* cKO mice compared with Ctrl mice (Appendix Fig. S5E,F). To further investigate the impact of *Setd8* depletion on NSC differentiation, we examined differentiation trends of YFP+ NSCs and found that the proportion of YFP+ NeuN+ mature neurons and YFP+ Dcx+ immature neurons was significantly reduced in *Setd8* cKO mice compared to Ctrl mice (Appendix Fig. S5G,K,L), indicating that neuronal differentiation was impaired in the absence of *Setd8*. Supporting this observation, *Setd8* cKO mice also exhibited a reduction in projections of YFP-positive neurons to the CA3 region (Appendix Fig. S5J). Additionally, a slight decrease in the proportion of astrocyte-differentiated cells was observed in *Setd8* cKO mice (Appendix Fig. S5H,I,M,N). Notably, a larger fraction of NSCs remained undifferentiated in *Setd8* cKO mice (Appendix Fig. S5H,O), suggesting that *Setd8* deletion impairs the transition from NSCs to differentiated neuronal and glial lineages. Since both NSC proliferation and neuronal differentiation were impaired, these factors likely contributed to the reduction in newly generated neurons. However, the rapid and prominent decrease in total NSC number observed one day after tamoxifen administration suggests that NSC loss cannot be fully explained by impaired proliferation alone. Therefore, we cannot exclude the possibility

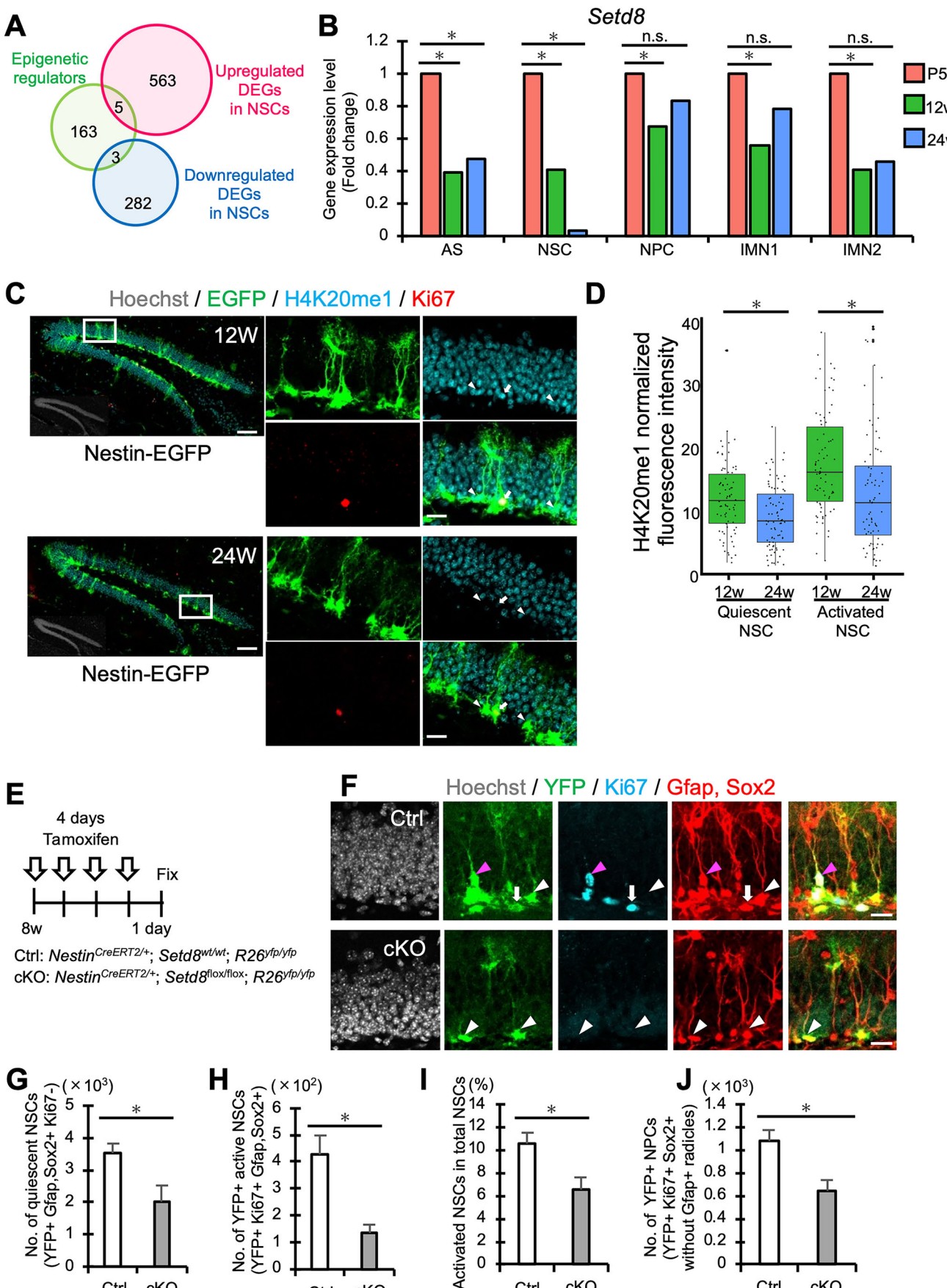

**Figure 3. Setd8 maintains NSC activity and neurogenesis in the DG.**

(A) Venn diagram showing overlap between epigenetic regulators and up- or downregulated DEGs in NSCs. (B) Bar graphs showing fold change of *Setd8* expression in each cluster compared to P5. *$q < 0.01$ by FindMarker in Seurat package with MAST compared with P5. n.s., not significant. Exact $p$ values from left to right: $p = 2.6618 \times 10^5$; $p = 0.0033$; $p = 2.2097 \times 10^5$; $p = 0.00019$; $p = 1.5876 \times 10^5$; $p = 0.6469$; $p = 1.1316 \times 10^6$; $p = 0.5275$; $p = 5.6776 \times 10^{11}$; $p = 0.0524$. (C) Representative images of staining for EGFP (green), H4K20me1 (cyan), Ki67(red), and Hoechst (gray; insets) in the hippocampal DG of Nestin-EGFP mice. The areas outlined by a white rectangle are enlarged to the right. Arrows and arrowheads indicate Ki67+ activated and Ki67− quiescent NSCs, respectively. Scale bars, 100 μm (left) and 20 μm (right). (D) Box plots showing H4K20me1 fluorescence intensity normalized by background in NSCs. Quiescent and active NSCs of the hippocampus in 24w mice displayed reduced signal intensity for H4K20me1 compared to the respective states of NSCs in 12w mice ($n = 76$ cells from three animals per group). *$p < 0.001$ by Wilcoxon rank-sum test. Box plots show the median (centre line), the 25th and 75th percentiles (bounds of box), and the minimum and maximum values (whiskers). Exact p values from left to right: $p = 0.0002829$; $p = 0.0007525$. (E) Experimental scheme for investigating Setd8 function in adult hippocampal NSCs using Ctrl (Nestin$^{CreERT2/+}$; *Setd8*$^{wt/wt}$; R26$^{yfp/yfp}$) and *Setd8* cKO mice (Nestin$^{CreERT2/+}$; *Setd8*$^{flox/flox}$; R26$^{yfp/yfp}$). (F) Representative images of staining for YFP (green), Ki67 (cyan), Gfap (red), Sox2 (red), and Hoechst (gray) of the hippocampus in Ctrl and *Setd8* cKO mice. White arrowheads, magenta arrowheads, and white arrows indicate quiescent NSCs, activated NSCs, and NPCs, respectively. Scale bars, 20 μm. (G, H) Quantification of the number of YFP+ Gfap/Sox2 + Ki67− quiescent NSCs (G) and YFP+ Gfap/Sox2 + Ki67+ active NSCs (H) in Ctrl and *Setd8* cKO ($n = 4$ animals per group). *$p = 0.036522487$ (G) and 0.016106224 (H) by Student's t test. (I) The proportion of YFP+ Gfap+ Sox2 + Ki67+ activated NSCs among YFP+ Gfap+ Sox2+ total NSCs in Ctrl and cKO ($n = 4$ animals per group). *$p = 0.04574894$ by Student's t test. (J) Quantification of the number of NPCs expressing YFP, Ki67, and Sox2 without Gfap+ radials in Ctrl and cKO ($n = 4$ animals per group). *$p = 0.026683966$ by Student's t test. Source data are available online for this figure.

that caspase-independent cell death also contributes to the depletion of NSCs.

Taken together, these results indicate that Setd8 maintains the proliferative and subsequent neurogenic capability of NSCs in the adult mouse hippocampus.

## Reduction of Setd8 expression leads to premature NSC depletion in the early life stage

LaminB1 loss disrupts nuclear envelope integrity and chromatin organization, leading to altered gene expression and activation of cellular senescence pathways (bin Imtiaz et al, 2021; Bedrosian et al, 2021). Prolonged (6.5 months) loss of LaminB1 has been reported to cause premature aging of mouse hippocampal NSCs in middle age (Bedrosian et al, 2021). These findings prompted us to investigate whether Setd8 might induce premature NSC aging when its expression is reduced over an extended period, starting at an early life stage (i.e., young adult stage). To address this, we utilized *Setd8* cKO mice and heterozygous *Setd8* conditional knockout (het-cKO; *Setd8*$^{flox/wt}$) mice carrying *Nestin-CreERT2* and *Rosa26-YFP* reporter transgenes. Ten-week-old (10w) cKO, het-cKO, and Ctrl mice were analyzed after 6 weeks of tamoxifen administration (Fig. 4A). We observed reductions in the number of both quiescent (Gfap+ Sox2+Ki67− or Nestin+ Sox2+Ki67−) and active (Gfap+ Sox2+Ki67+ or Nestin+ Sox2+Ki67+) NSCs in *Setd8* cKO mice compared to the Ctrl (Fig. 4B–F; Appendix Fig. S6A). The proportion of active NSCs among total NSCs was also lower in the DG of 10w *Setd8* cKO mice (Fig. 4G). The extent of these declines in 10-week-old *Setd8* cKO mice is similar to that in 30-week-old control mice (Appendix Fig. S6B–D), indicating an accelerated loss of NSCs in cKO mice. The het-cKO mice showed a reduced number of NSCs in line with the decreased activation of NSCs compared to Ctrl mice, although not as much as in *Setd8* cKO mice (Fig. 4B–F; Appendix Fig. S6A), probably due to the dosage of the gene. These data suggest that the downregulation of *Setd8* causes severe dysfunction and premature aging of NSCs at the young adult stage.

Decreased neurogenesis related to aging leads to a decline in hippocampal memory function (Enwere et al, 2004; Jessberger and Gage, 2008; McAvoy et al, 2016). Therefore, we next conducted a hippocampus-dependent memory test (Goodman et al, 2010), the

novel place recognition test (NPRT) (Fig. 4H). None of the mice showed a strong interest in a particular object during training (Fig. 4I). *Setd8* cKO and het-cKO mice showed a reduced preference for an object placed at the novel location compared with Ctrl (Fig. 4I), whereas locomotor activity was not different among these groups (Appendix Fig. S6E–G). We also conducted another hippocampus-related memory test, the contextual fear conditioning test. *Setd8* cKO and het-cKO mice displayed significantly decreased contextual fear memory compared with Ctrl (Appendix Fig. S6H,I). These data indicate that hippocampus-dependent memory function is impaired in the *Setd8* expression-reduced mice in parallel with impaired NSC activity at the young adult stage.

## Downregulation of Setd8 establishes aging-related molecular signatures in NSCs

To gain deeper insight into how the reduction of *Setd8* expression shapes NSC aging, we analyzed the transcriptome of cultured mouse NSCs with lentivirus-mediated *Setd8* KD after selection with puromycin (Fig. 5A). *Setd8* expression was indeed reduced by *Setd8* shRNA at three days after lentiviral infection (Appendix Fig. S7A). We identified genes whose expression was either upregulated ($n = 995$) or downregulated ($n = 1049$) in *Setd8* KD cells relative to Ctrl (fold change ≥ 1.5, q < 0.05) (Fig. 5B). GO analysis showed that genes suppressed by *Setd8* KD were enriched in processes associated with cell cycle regulation (Fig. 5C,D), consistent with our findings that *Setd8* KD or deletion suppressed NSC proliferation in vitro and in vivo (Figs. 3I and 4G; Appendix Fig. S4B–H). Furthermore, gene set enrichment analysis (GSEA) revealed that *Setd8* KD significantly increased and decreased the expression of the upregulated and downregulated DEGs, respectively, in hippocampal NSCs with age depicted in Fig. 1 (Fig. 5E,F). We also checked the enrichment of *Setd8*-KD-induced DEGs in age-related gene expression changes in NSCs (Fig. 1E) compared to a random gene set (Fig. 5G). We found that genes downregulated by *Setd8* KD significantly overlap with genes downregulated in NSCs with aging, but not with genes upregulated. On the other hand, genes upregulated by *Setd8* KD significantly overlapped with genes upregulated in NSCs with aging, and to a lesser extent with genes downregulated. These results suggest that Setd8 KD induces gene

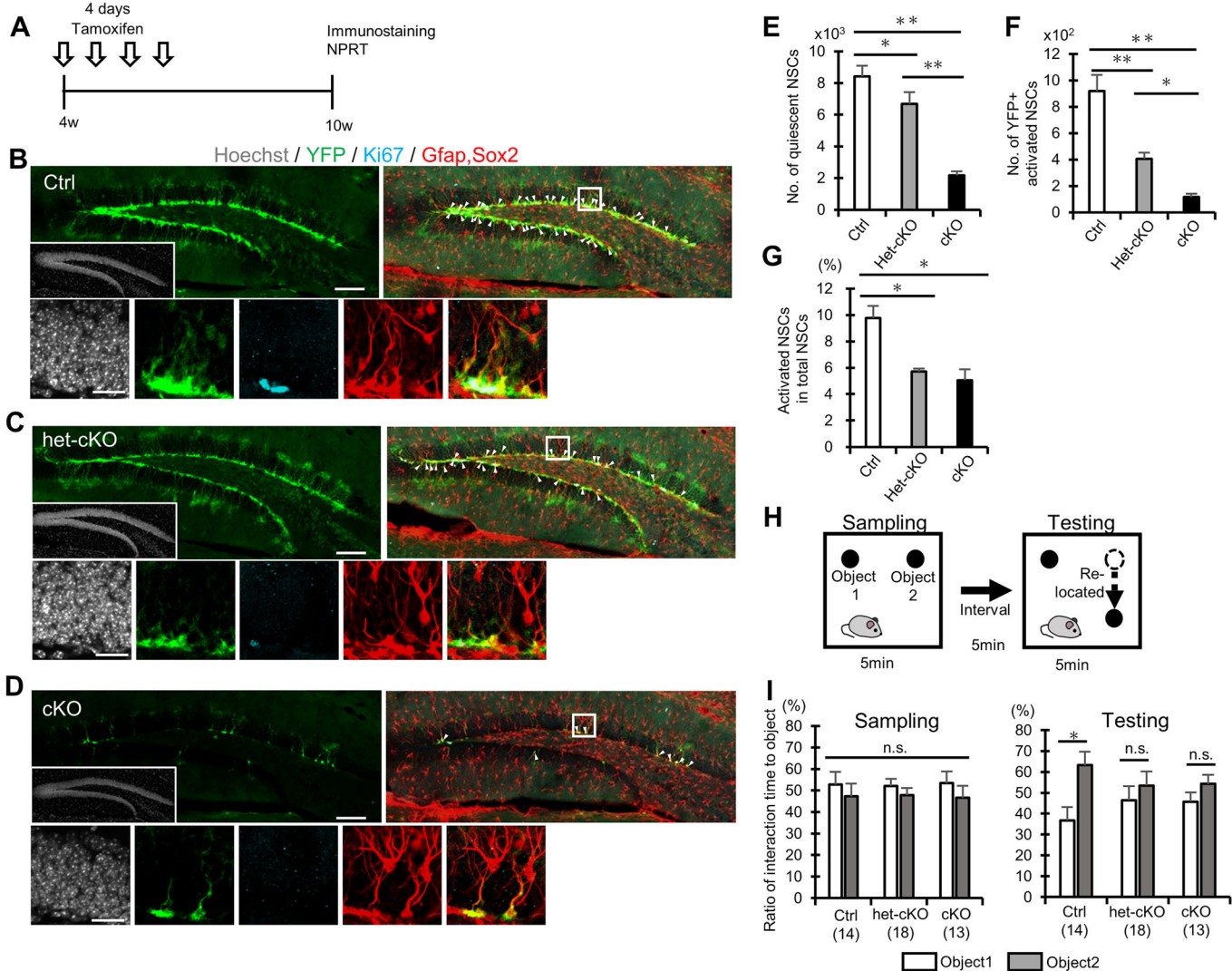

**Figure 4.   Reduction of Setd8 expression causes premature depletion of NSCs and memory impairment.**

(A) Experimental scheme for assessing the effect of *Setd8* expression decrease on NSCs and neurogenesis in the DG at young adult age (10w). (B–D) Representative images of staining for YFP (green), Ki67 (cyan), Gfap (red), Sox2 (red), and Hoechst (gray; insets) of the DGs in Ctrl (B) ($n = 4$ animals), *Setd8* het-cKO (C) ($n = 4$ animals), and *Setd8* cKO (D) ($n = 4$ animals). Arrowheads indicate YFP-labeled NSCs. Lower panels show higher magnification of upper panel. Scale bars, 100 μm (upper) and 20 μm (lower). (E, F) Quantification of the number of YFP+ Gfap/Sox2 + Ki67− quiescent (E) and YFP+ Gfap/Sox2 + Ki67+ activated (F) NSCs in Ctrl, het-cKO, and cKO ($n = 4$ animals per group). *$p < 0.05$ and **$p < 0.01$ by ANOVA with Tukey post hoc tests. Statistical data are presented as mean ± SEM. Exact $p$ values from left to right: $p = 0.0489931$ (E); $p = 0.0012056$ (E); $p = 0.0001080$ (E); $p = 0.0030571$ (F); $p = 0.00658989$ (F); $p = 0.0001245$ (F). (G) Proportion of YFP+ Gfap+ Sox2 + Ki67+ activated NSCs among YFP+ Gfap+ Sox2+ total NSCs in Ctrl, het-cKO, and cKO ($n = 4$ animals per group). *$p < 0.01$ by ANOVA with Tukey post hoc tests. Statistical data are presented as mean ± SEM. Exact $p$ values from left to right: $p = 0.0083019$; $p = 0.7975168$; $p = 0.0032564$. (H) Schematic diagram of the NPRT. (I) Exploration ratio for time spent in exploring the object during sampling phase (left) and the relocated object during the test phase (right) in Ctrl, het-cKO, and cKO. het-cKO and cKO mice lacked a preference for the relocated object. *$p < 0.05$ by Student's t test. n.s., not significant. $n = 14$ (Ctrl), 18 (het-cKO) and 13 (cKO). Statistical data are presented as mean ± SEM. Exact $p$-values from left to right are as follows: for sampling, $p = 0.3243$, $p = 0.2591$, and $p = 0.2734$; for testing, $p = 0.0303$, $p = 0.3020$, and $p = 0.1761$. Source data are available online for this figure.

expression changes that resemble those observed in aging NSCs, indicating that at least some age-dependent changes in gene expression in hippocampal NSCs are mediated by decreased Setd8 expression.

We and others have shown that NSCs enter a deep quiescent state and become less active with age (Appendix Fig. S1O–Q) (Harris et al, 2021). Next, we examined whether *Setd8* KD increased the expression of genes associated with quiescence, and found that the DEGs with increased or decreased expression in cultured NSCs

were enriched for genes associated with quiescent or active NSC genes, respectively (Fig. 5H; Appendix Fig. S7B,C). For instance, *Id4* and *Notch2*, which were shown previously to be involved in cellular quiescence (Blomfield et al, 2019; Zhang et al, 2019b) are upregulated (Fig. 5I,J; Appendix Fig. S7D–F). The hallmark gene for deep quiescence, *Apoe* (Harris et al, 2021), was also upregulated by *Setd8* KD (Fig. 5K). Additionally, *Setd8* KD reduced *Ascl1* and *Cdk1* expression (Fig. 5L,M), which is associated with NSC activation (Beck et al, 2012; Andersen et al, 2014). These data

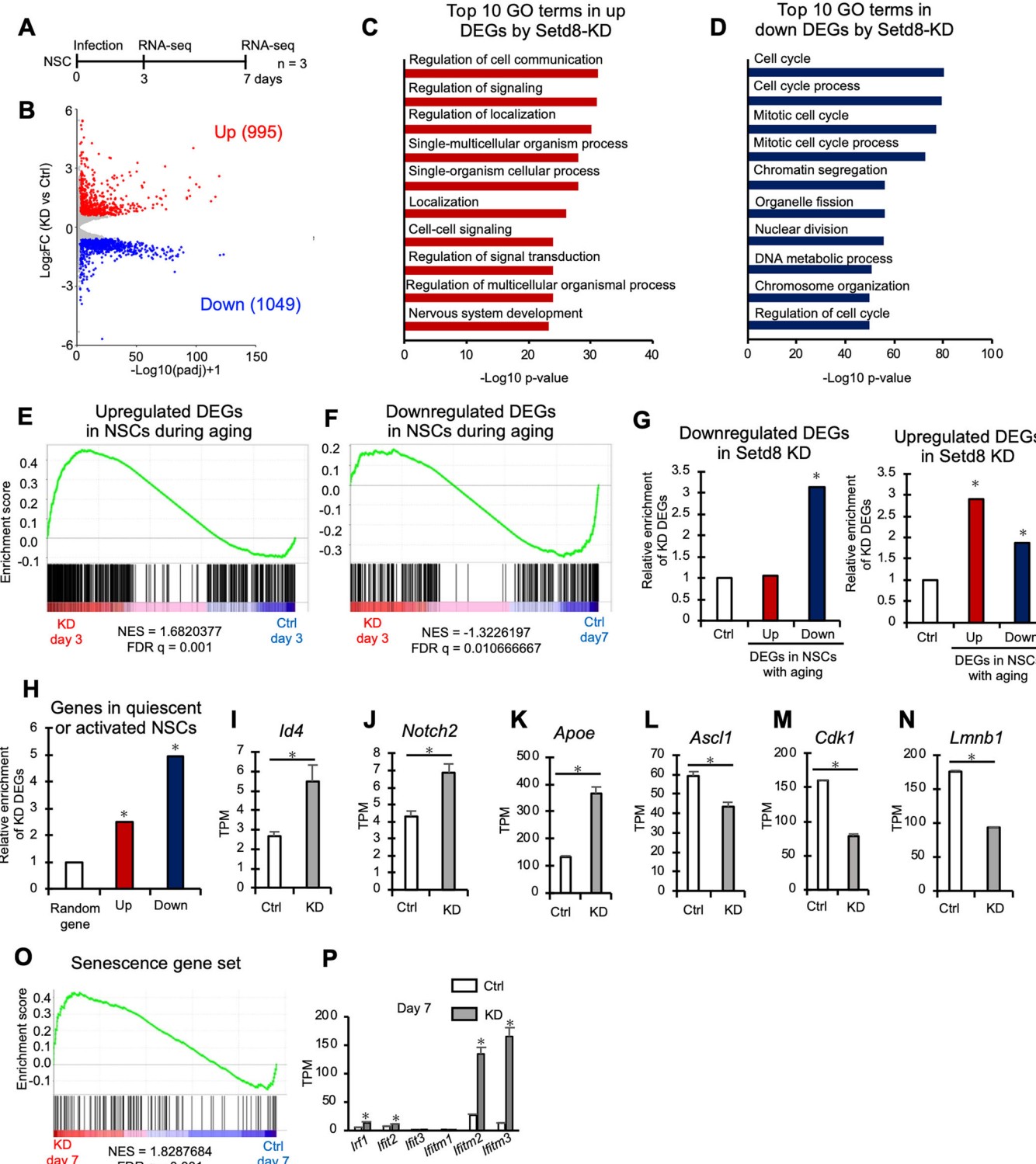

suggest that reducing *Setd8* expression reinforces quiescence in NSCs by increasing the expression of quiescence-associated genes.

We then asked whether the reduction of *Setd8* expression alters cellular senescence-associated gene expression. The expression of *Apoe* was increased by *Setd8* KD, whereas *Lmnb1* expression was reduced (Fig. 5K,N; Appendix Fig. S7D–F), at three days after

lentiviral infection, indicating that senescence-associated gene expression changes within a short period after both *Setd8*-KD and -KO. To determine the effects of relatively long-term suppression of *Setd8* expression on cellular senescence in NSCs, we analyzed the transcriptome of cultured NSCs at seven days after lentiviral infection. GSEA revealed that long-term *Setd8* KD

**Figure 5.   Molecular signature of NSC aging is established by Setd8 downregulation.**

(A) Experimental scheme for investigating transcriptomic alteration in NSCs in vitro by RNA-seq ($n = 3$). (B) Scatter plot of genes expressed in NSCs from Ctrl and *Setd8*-KD mice. Upregulated (red) and downregulated (blue) DEGs in *Setd8*-KD (q value < 0.05 by DESeq2 package, fold change ≥ 1.5) are highlighted. (C, D) The top 10 GO terms in significantly up- ($n = 995$) (C) or downregulated ($n = 1049$) (D) DEGs in *Setd8*-KD cells. (E, F) GSEA showing the significant enrichment of up- (E) or downregulated (F) DEGs of in vivo NSCs with age (identified in Fig. 1E) in *Setd8*-KD cells or Ctrl cells, respectively. (G) Enrichment analysis of up- (red) or downregulated (blue) DEGs of NSCs with age in vivo, identified in Fig. 1E, on *Setd8*-KD-induced up- or downregulated genes. The leftmost bar represents the expected overlap based on a random gene set, while the remaining bars show the fold enrichment of actual overlaps. *p < 0.01 by hypergeometric distribution. Exact *p* values from left to right: $p = 2.52 \times 10^{-18}$; $p = 3.48 \times 10^{-15}$; $p = 2.59 \times 10^{-10}$. (H) Enrichment analysis of NSC quiescence- (red) or activation-associated (blue) genes, reported by Shin et al, 2015, on *Setd8*-KD-induced up- or downregulated genes. The leftmost bar represents the expected overlap based on a random gene set, while the remaining bars show the fold enrichment of actual overlaps. *p < 0.001 by hypergeometric distribution. Exact *p* values from left to right: $p = 3.31 \times 10^{-18}$; $p = 1.17 \times 10^{-91}$. (I–N) Bar graphs showing expression levels of the indicated genes in Ctrl (white) and KD (gray) NSCs. *q < 0.01 by DESeq2 package. $n = 3$ samples. Statistical data are presented as mean ± SEM. (O) GSEA results showing significant enrichment of the senescence-associated gene set, as reported by Saul et al, 2022, in Ctrl (white) and KD (gray) NSCs after long-term suppression of *Setd8*, observed 7 days post-infection. (P) Expression level of interferon signaling-related genes in Ctrl (white) and KD (gray) NSCs after long-term suppression of *Setd8*, observed 7 days post-infection. *q < 0.01 by DESeq2 package. $n = 3$ samples. Statistical data are presented as mean ± SEM. Exact p values from left to right: $q = 0.0167630229897316$; $q = 0.0167$; $q = 1.43528482781079 \times 10^{-11}$; $q = 3.495 \times 10^{-51}$.

expanded the senescence gene set (SenMayo), which was defined in a previous study (Fig. 5O) (Saul et al, 2022). Moreover, *Setd8* KD increases interferon signaling-related gene expression, similar to age-related changes in in vivo NSCs (Fig. 5P). These data indicate that declining *Setd8* expression promotes cellular senescence-related gene expression. Taking these findings together, decreased expression of *Setd8* mediates at least a fraction of the age-dependent changes in gene expression, such as those associated with quiescence and senescence, in hippocampal NSCs, resulting in reduced NSC activity.

To further investigate the downstream effects of *Setd8* downregulation, we checked expression of epigenetic gene using our RNA-seq data and found that *Setd8* downregulation in NSCs led to significant changes in the expression of multiple epigenetic regulators, including *Tet1*, *Tet2*, and several *HDAC* family genes (Appendix Fig. S7G, Dataset EV4 and EV5). These results suggest that Setd8 functions in coordination with other epigenetic regulators and that, beyond its role in H4K20me1 modification, Setd8 contributes to broader epigenomic regulation, influencing chromatin dynamics and cell proliferation in accordance with previous reports (Oda et al, 2009; Shoaib et al, 2021).

## Downregulation of Setd8 induces age-related epigenomic alteration

To take a closer look at transcriptomic alteration induced by the *Setd8* reduction in NSCs, we decided to focus on H4K20me1 modification and chromatin state. Samples collected from cultured mouse NSCs in *Setd8* KD after selection with puromycin were subjected to ChIP-seq (H4K20me1) and ATAC-seq (Fig. 6A). Since H4K20me1 is enriched in the gene body (Tanaka et al, 2017), we calculated the enrichment of H4K20me1 within the gene body of all genes and observed that *Setd8* KD caused the reduction of H4K20me1 levels there (Fig. 6B), consistent with immunostaining results showing a reduced signal intensity of H4K20me1 (Appendix Fig. S8A,B). The level of H4K20me1 was lower around the loci of downregulated genes induced by *Setd8* KD (Fig. 6C), whereas no significant difference was observed around upregulated gene loci (Appendix Fig. S8C). Moreover, genes located in genomic regions where H4K20me1 modification levels in the gene body decreased by more than 2-fold in *Setd8* KD NSCs significantly overlapped with downregulated DEGs in *Setd8* KD NSCs (Fig. 6D). These data suggest that *Setd8* downregulation inhibits gene expression by

reducing H4K20me1 levels in the gene body, whereas H4K20me1 within the gene body may not be involved in *Setd8* downregulation-induced increased gene expression.

To examine the effect of reduced *Setd8* expression and subsequent H4K20me1 reduction on chromatin accessibility, we performed ATAC-seq of cultured NSCs with *Setd8* KD after puromycin selection. Peak calling with MACS2 identified approximately 80,000 open chromatin regions (open regions in Ctrl plus those in *Setd8* KD NSCs) (Fig. 6E). There were more gained-closed DARs (31,689 regions) than gained-open DARs (4191 regions) in cells with reduced *Setd8* expression, similar to the trend of a 10-fold increase in gain-closed DARs in in vivo NSCs with age (Figs. 2D and 6E). We then assessed chromatin accessibility near up- and downregulated DEGs in *Setd8* KD NSCs and observed increased and decreased accessibility, respectively (Fig. 6F), suggesting that chromatin accessibility changes are concordant with the corresponding gene expression changes in vitro. Moreover, motif enrichment analysis revealed that retinoic acid-related orphan receptors (Rora, Rorb, and Rorc) were enriched in gained-opened DARs induced by *Setd8* KD, whereases multiple NSC and NPC activity-regulating transcription factors, such as Ascl1, Nfix, Zeb1, Neurod1, and Neurog2, were enriched in gained-closed DARs (Fig. 6G), similar to what was observed in in vivo NSCs with age, implying that decreased expression of *Setd8* triggers chromatin accessibility changes that occur in NSCs and NPCs with age.

We then investigated whether decreased *Setd8* expression alters accessible chromatin regions that change with aging in NSCs in vivo. Chromatin accessibility of age-related gained-closed DARs in NSCs in vivo, defined in Fig. 2, was significantly reduced by *Setd8* KD in cultured NSCs, whereas age-related gained-open DARs showed almost no change (Figs. 2D,E and 6H). Moreover, *Setd8* KD resulted in a significant decrease in H4K20me1 level in age-related gained-closed DARs in in vivo NSCs (Fig. 6I). These changes are also clearly evident at the level of individual genes (Fig. 6J,K). However, some genes, such as *Apoe*, showed almost no changes in chromatin accessibility and H4K20me1 levels even when their expression levels were altered in NSCs by *Setd8* KD (Figs. 5K and 6L). Nevertheless, our findings suggest that the downregulation of *Setd8* mediates changes in the majority of age-dependent gene expression in hippocampal NSCs by altering chromatin accessibility and H4K20me1 level.

Finally, we investigated whether the proliferation of NSCs, which had been reduced due to Setd8 dysfunction, could be

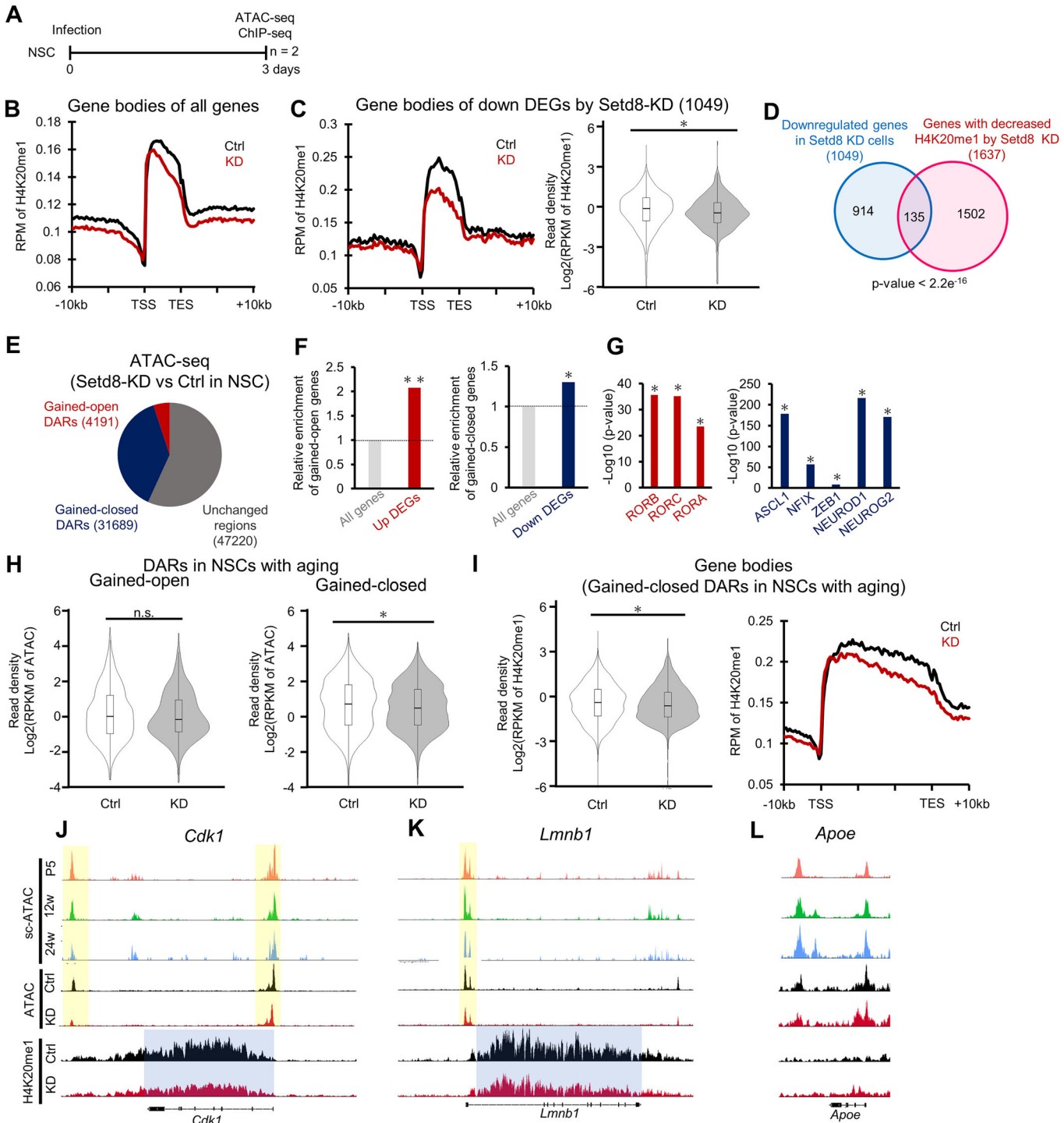

**A** Infection: NSC at day 0. ATAC-seq ChIP-seq at 3 days, n = 2

**B** Gene bodies of all genes

**C** Gene bodies of down DEGs by Setd8-KD (1049)

**D** Downregulated genes in Setd8 KD cells (1049): 914. Genes with decreased H4K20me1 by Setd8 KD (1637): 1502. Overlap: 135. p-value < 2.2e⁻¹⁶

**E** ATAC-seq (Setd8-KD vs Ctrl in NSC). Gained-open DARs (4191), Gained-closed DARs (31689), Unchanged regions (47220)

**F** Relative enrichment of gained-open genes; Relative enrichment of gained-closed genes

**G** −Log10 (p-value): RORB, RORC, RORA; ASCL1, NFIX, ZEB1, NEUROD1, NEUROG2

**H** DARs in NSCs with aging. Gained-open; Gained-closed

**I** Gene bodies (Gained-closed DARs in NSCs with aging)

**J** Cdk1 **K** Lmnb1 **L** Apoe

restored. Since the long-term *Setd8* downregulation in NSCs may have caused nearly irreversible changes in chromatin modification, likely due to cellular senescence, we conducted experiments using the selective Setd8 inhibitor UNC0379 (Tanaka et al, 2017) that can immediately restore Setd8 activity without altering its protein levels (Appendix Fig. S9A). Treatment with the inhibitor effectively reduced H4K20me1 levels and suppressed NSC proliferation.

Importantly, upon inhibitor removal, H4K20me1 levels returned to normal, and NSC proliferation was restored (Appendix Fig. S9B–F). These data suggest that the reduction in NSC proliferation caused by a transient reduction in Setd8 activity is reversible once Setd8 function is restored. Therefore, the early stages of aging, when *Setd8* expression begins to decline, may represent a critical window for intervention.

**Figure 6.   Downregulation of Setd8 induces aging-related epigenomic alteration.**

(A) Experimental scheme for investigating epigenetic alteration in NSCs in vitro by ChIP-seq for H4K20me1 ($n = 2$) and ATAC-seq ($n = 2$). (B) Enrichment profile of H4K20me1 in gene body of all genes in Ctrl (black) and KD (red) NSCs. (C) Enrichment profile (left) and violin plot (right) of H4K20me1 around gene bodies of downregulated DEGs ($n = 1049$ genes) by *Setd8*-KD in Ctrl (black) and KD (red) (left) NSCs. *$p = 2.371 \times 10^{-6}$ by Wilcoxon rank-sum test. Box plots show the median (centre line), the 25th and 75th percentiles (bounds of box), and the minimum and maximum values (whiskers). (D) Venn diagram showing significant overlap between the downregulated DEGs by *Setd8*-KD and genes with decreased H4K20me1 in the gene body (Fisher's test). (E) Pie chart showing the percentage of accessible and inaccessible regions of *Setd8*-KD NSCs. (F) Enrichment analysis of genes associated with gained-open or -closed DARs within 10 kb up- or downstream of a TSS, on *Setd8* KD-induced up- (left) or downregulated genes (right). *$p < 0.001$ and **$p < 0.0001$ by hypergeometric distribution. (G) Bar graphs showing the significant enrichment of motifs associated with NSCs and NPC activity in the gained-open (left) and -closed DARs (right) in *Setd8*-KD NSCs. *$p < 0.01$ by hypergeometric distribution. (H) Violin plots showing chromatin accessibility of gained-open (left) or -closed DARs (right) of in vivo NSCs with age between Ctrl and *Setd8*-KD cells. *$p < 0.001$ by Wilcoxon rank-sum test. n.s., not significant. Violin plots show the median (centre line), the 25th and 75th percentiles (bounds of box), and the minimum and maximum values (whiskers). $n = 751$ (gained-open) and 7488 (gained-closed). Gained-open: $p = 0.3082$; Gained-closed: $p = 8.61 \times 10^{-10}$. (I) Violin plot (left) and enrichment profile (right) at gained-closed DARs of in vivo NSCs with age for H4K20me1 in Ctrl and *Setd8*-KD cells. *$p = 8.34 \times 10^{-5}$ by Wilcoxon rank-sum test. Box plots show the median (centre line), the 25th and 75th percentiles (bounds of box), and the minimum and maximum values (whiskers). $n = 7488$ (gained-closed). (J–L) Alignment data for scATAC-seq in NSCs in vivo (P5, 12w, and 24w), ATAC-seq in NSCs in vitro (*Setd8*-KD and Ctrl), and ChIP-seq to H4K20me1 in NSCs in vitro (*Setd8*-KD and Ctrl) at the *Cdk1*, *Lmnb1*, and *Apoe* loci. The regions identified by peak-calling are highlighted in yellow, and the gene bodies are highlighted in light blue.

## Discussion

In the present study, we have shown that NSCs exhibit an early onset of aging in the hippocampus, characterized by altered chromatin accessibility in the regulatory factor-binding regions of genes involved in stemness and neurogenesis with corresponding transcriptional changes. *Setd8* downregulation causes NSC aging and age-related hippocampal dysfunction through at least a proportion of the age-related changes in gene expression and epigenetics. Thus, we have uncovered a novel mechanism for how cell-intrinsic molecular changes, accompanied by epigenetic changes, elicit stem cell aging in the mammalian brain during the early stages of chronological aging.

NSC activity and neurogenesis are well known to decline with age. Previous studies have compared transcriptomes and epigenomes between young and old quiescent NSCs (Kalamakis et al, 2019; Leeman et al, 2018; Yeo et al, 2023; Buckley et al, 2023; Li et al, 2024). However, such comparisons have thus far not revealed molecular aging of NSCs at very early stages of chronological aging. Another recent study addressed transcriptomic changes of NSCs at different chronological ages, although it still remains largely unclear when and how transcriptional changes linked to epigenetic remodeling of NSCs begin to occur (Ibrayeva et al, 2021; Harris et al, 2021). In addition, it is not well understood whether cells other than NSCs that are involved in neurogenic processes, such as NPCs and IMNs, change their transcriptome and epigenome with age. Therefore, in this study, we analyzed the transcriptomic and epigenomic alterations of NSCs as well as their progeny with age by leveraging scRNA-seq and scATAC-seq analyses. We found that NSCs from the mature brain have already undergone changes in gene expression related to cellular quiescence and senescence, such as inflammatory signaling, consistent with previous reports (Ibrayeva et al, 2021; Harris et al, 2021), indicating that NSCs exhibit early aging in the mature hippocampus. Furthermore, integrative analyses of scRNA-seq and scATAC-seq revealed that the chromatin landscape of hippocampal NSCs, as well as their progeny, including NPCs and IMNs, changed continuously from the neonatal to the mature adult stage, with corresponding transcriptional changes. This is probably because age-related transcriptomic and epigenomic changes in the mother cells, the NSCs, are to some extent inherited by their progeny. Moreover, the fact that the gene expression profiles of newly generated neurons

were all different at P5, 12w, and 24w suggested an age-specific role that has rarely been considered. Thus, we propose that changes in the properties of both NSCs and their progeny during the early stages of chronological aging are responsible for disrupting the neurogenic process with age and provide a resource to investigate the associated molecular mechanisms at the systems level.

Both dormant cellular states, i.e., quiescence and senescence, are traditionally considered distinct. Quiescence is reversible to a proliferative state upon growth stimulation, whereas senescence is irreversible in physiological conditions. Cellular quiescence is not homogeneous but rather a heterogeneous state represented by a shallow or deep quiescence state (Coller et al, 2006). For instance, as seen in aging NSCs, cells may enter a deeper dormant state (Appendix Fig. S1R,S) (Ibrayeva et al, 2021; Harris et al, 2021), and this dormancy, although reversible, takes longer than in younger NSCs to return to the cell cycle (Harris et al, 2021). Recent findings suggest that quiescence deepening is a common transition path toward senescence in many cell and tissue types. Prolonged inhibition of the lysosomal autophagy function in quiescent embryonic fibroblasts has been reported to eventually push cells into a senescence-like state, characterized by an irreversibly arrested phenotype with β-galactosidase activity and hypertrophy (Fujimaki et al, 2019). Gene expression profiles have also suggested a similarity between deep quiescence and senescence states. A core set of senescence-associated genes identified in a meta-analysis (Hernandez-Segura et al, 2017) were similarly down- or upregulated in quiescence deepening as well as in senescence (Fujimaki et al, 2019). Moreover, quiescence-associated gene sets were also altered in a broad array of senescent and aging cell types in vitro and in vivo (Fujimaki et al, 2019; Tümpel and Rudolph, 2019; Kovtonyuk et al, 2019; Sousa-Victor et al, 2014). Thus, while quiescence can generally be considered to be distinct from senescence, quiescence deepening appears to reflect a common transitional path, with gradually reduced proliferative tendencies leading to irreversible senescence, related to the aging process. We discovered that NSCs in the mature adult stage already exhibit changes in gene expression that are associated with cellular quiescence and senescence, and that some of these genes also contain gene sets commonly associated with both quiescence and senescence, such as *ApoE* (Artegiani et al, 2017; Zhao et al, 2022), although it is not clear whether cells at this stage are in the process of transitioning or have already transitioned to a senescent state.

Further investigation is therefore warranted to reveal whether NSCs adversely affect surrounding cells by releasing senescence-associated secretory phenotype factors as do senescent cells in the mature adult brain.

During development, embryonic neural precursor populations generate first neurons and then glial cells by sequentially acquiring the epigenetic potential to differentiate into neurons and glial cells (Takizawa et al, 2001; Namihira et al, 2009; Katada et al, 2021). A recent study has reported an embryonic origin for adult neural progenitors in the mouse hippocampus and also suggested a continuous model in which a single precursor population contributes continuously and exclusively to dentate neurogenesis from early embryonic stages to adulthood (Berg et al, 2019). We found that hippocampal NSCs do not initially possess the appropriate chromatin state to differentiate into neurons, but rather acquire it as the neurogenic process progresses prior to neuronal gene expression. Therefore, these observations suggest that neural precursor populations that have not acquired neurogenic chromatin status from embryonic to adult stages behave as adult NSCs in the hippocampus. Furthermore, we showed that the expression of specific marker genes for hippocampal NSCs remains relatively constant with age. Nonetheless, we also observed age-related changes in gene expression and chromatin accessibility in NSCs. Thus, it is conceivable that although the fundamental identity of hippocampal NSCs is maintained during the fetal and postnatal aging process, these age-dependent changes gradually impair NSC functions.

We have demonstrated that *Setd8* downregulation causes NSC aging and age-related hippocampal dysfunction through at least a proportion of the age-related changes in gene expression and in epigenetic states (H4K20me1 and chromatin accessibility changes). Lamins have previously been implicated in aging and NSC biology (de Leeuw et al, 2018; Shimi et al, 2010; bin Imtiaz et al, 2021; Bedrosian et al, 2021). Although prolonged loss (6.5 months) of LaminB1 caused premature aging of mouse hippocampal NSCs in a previous study (Bedrosian et al, 2021), we found that the loss of *Setd8* for six weeks induces premature NSC aging. This dramatic effect probably arises because Setd8 is upstream of genes such as *LaminB1* and *Ascl1* that decline with age and play critical roles in NSC functions. Furthermore, since it is upstream of these genes, Setd8 is likely to be a factor that acts early in aging and/or senescence. We also found that Setd8 could cause expression change in defined age-related gene sets without altered chromatin accessibility or H4K20me1 level. Recently, non-histone proteins, such as p53, PCNA, Numb, and UHRF1, have been reported as substrates of Setd8 and to have monomethylated lysine residues, which is widely recognized as a fundamental post-translational modification (Huen et al, 2008; Takawa et al, 2012; Shi et al, 2007; Dhami et al, 2013; Zhang et al, 2019a). Other previous studies have shown that p53 methylation by Setd8 decreases tumor suppressor activity (Herviou et al, 2021; Kukita et al, 2022; Veschi et al, 2017). These findings suggest that Setd8 regulates NSC behavior not only through histone monomethylation but also through monomethylation of such non-histone proteins, which may alter the expression of age-related genes without changing chromatin accessibility. Our research provides a resource for uncovering the ever-changing process of neurogenesis with age and sheds light on epigenetic alteration to understand the mechanism underlying the aging signature associated with adult hippocampal NSCs during the early stages of aging. Thus, we believe that our findings will hasten the discovery of new therapeutic targets for age-related functional decline of the brain.

# Methods

**Reagents and tools table**

| Reagent/Resource | Reference or Source | Identifier or Catalog Number |
|---|---|---|
| **Experimental models** | | |
| Nestin-EGFP | Yamaguchi et al, 2000 | |
| Setd8 fl/fl | Oda et al, 2009 | |
| Nestin-CreERT2 | Imayoshi et al, 2006 | |
| Rosa26-YFP | F. Costantini | |
| **Recombinant DNA** | | |
| pLLX vector | Drs. Z. Zhou and M. E. Greenberg | |
| FUIGW plasmid | Drs. Z. Zhou and M. E. Greenberg | |
| LentiCRISPR v2 vector | Addgene | 52961 |
| Mouse CRISPR Knockout Pooled Library (Brie) | Addgene | 73633 |
| **Antibodies** | | |
| Chicken anti-Nestin | Aves Labs | NES |
| Mouse anti-Ki67 | BD Biosciences | 550609 |
| Rat anti-Ki67 | eBioscience | 14-5698-84 |
| Chicken anti-Gfap | Millipore | AB5541 |
| Goat anti-Sox2 | Santa Cruz Biotechnology | sc-17320 |

| Reagent/Resource | Reference or Source | Identifier or Catalog Number |
|---|---|---|
| Goat anti-DCX | Santa Cruz Biotechnology | sc-8066 |
| Rabbit anti-DCX | Cell Signaling Technology | 4604 |
| Mouse anti-NeuN | Millipore | MAB-377 |
| Rabbit anti-active Caspase 3 | R&D Systems | AF-835 |
| Chicken anti-GFP | Aves Labs | GFP-1010 |
| Rat anti-GFP | Nacalai Tesque | 04404-26 |
| Rabbit anti-H4K20me1 | Abcam | ab9051 |
| Rat anti-BrdU | Abcam | ab-6326 |
| CF-488 donkey anti-mouse IgG | Biotium | 20014 |
| CF-488 donkey anti-chicken IgG | Biotium | 20166 |
| CF-555 donkey anti-rabbit IgG | Biotium | 20038 |
| CF-555 donkey anti- mouse IgG | Biotium | 20037 |
| CF-568 donkey anti-rat IgG | Biotium | 20092 |
| CF-647 donkey anti-rabbit IgG | Biotium | 20047 |
| CF-647 donkey anti-goat IgG | Biotium | 20048 |
| **Oligonucleotides and other sequence-based reagents** | **Sequence** | |
| Setd8 fl/fl forward primer | 5′-TGT AAG AGA ACT TTG AAT GG-3′ | |
| Setd8 fl/fl reverse primer | 5′-AGG CAG GGG GAG GAT-3′ | |
| Rosa26-YFP 1st primer | 5′-AAA GTC GCT CTG AGT TGT TAT-3′ | |
| Rosa26-YFP 2nd primer | 5′-GCG AAG AGT TTG TCC TCA ACC-3′ | |
| Rosa26-YFP 3rd primer | 5′-GGA GCG GGA GAA ATG GAT ATG-3′ | |
| Setd8 KD_1-Foward | 5′-tGGGAGACTTTGTGGTAGAATAttcaagagaTATTCTACCACAAAGTCTCCCtttttttggaac-3′ | |
| Setd8 KD_1-Reverse | 5′-tcgagttccaaaaaaGGGAGACTTTGTGGTAGAATAtctct tgaaTATTCTACCACAAAG TCTCCCa-3′ | |
| Setd8 KD_2-Foward | 5′-tGCCTCCTTC AAAGGACAAAGTttcaagagaACTTTGTCCTTTGAAGGAGGCtttttttggaac-3′ | |
| Setd8 KD_2-Reverse | 5′-tcgagttccaaaaaaGCCTCCTTCAAAGGACAAAGTtctcttgaaACTTT GTCCTTTGAAGGAGGCa-3′ | |
| Setd8_gRNA1 | GAACTCGGTTGCCCATCATG | |
| Setd8_gRNA2 | CTGCTTACCCCGTCGCTGCG | |
| Setd8_gRNA3 | TCACAGATTTCTACCCTGTG | |
| Setd8_gRNA4 | AGCCCTGAAAAAGTCCCTCA | |
| Ctrl_gRNA1 | AAAAAGTCCGCGATTACGTC | |
| Ctrl_gRNA2 | AACGGTAGCGTACCCGTGAA | |
| Ctrl_gRNA3 | CCATCGGTTCGACTTACCGC | |
| Ctrl_gRNA4 | TAAGACTGGGTGTCCCGCGT | |
| **Chemicals, Enzymes and other reagents** | | |
| EdU | Wako | E915020 |
| Tamoxifen | Sigma-Aldrich | T5648 |
| Sesame oil | Nacalai Tesque | 25620-65 |
| Paraformaldehyde (PFA) | Sigma-Aldrich | P6148 |
| Sucrose | Nacalai Tesque | 30404-45 |
| OCT compound | Tissue Tek, Sakura Finetek | 25608-930 |
| Papain | Nacalai Tesque | 26036-34 |
| Minimum Essential Medium α | Gibco | 12571063 |
| BSA | Sigma-Aldrich | A9418 |
| Hanks' balanced salt solution (HBSS) | Gibco | 14025092 |

| Reagent/Resource | Reference or Source | Identifier or Catalog Number |
|---|---|---|
| DNase I | Invitrogen | 18068015 |
| 7-AAD | BD Pharmingen | 559925 |
| Fetal bovine serum (FBS) | Invitrogen | 10270106 |
| Triton X-100 | Nacalai Tesque | 35501-02 |
| Hoechst 33258 | Nacalai Tesque | 19172-51 |
| Poly-L-ornithine | Sigma-Aldrich | P3655 |
| Fibronectin | Sigma-Aldrich | F1141-5MG |
| Dulbecco's modified Eagle's medium (DMEM)/F-12 | Gibco | D8900 |
| N2 supplement | Gemini | 400-163 |
| Basic fibroblast growth factor (bFGF) | PeproTech | 100-18B |
| Epidermal growth factor (EGF) | PeproTech | AF-100-15 |
| Penicillin/streptomycin/fungizone | Nacalai Tesque | 26253-84 |
| DMSO | Nacalai Tesque | 13445-74 |
| Puromycin | Sigma-Aldrich | P8833 |
| UNC0379 | Selleck | S8572 |
| Dynabeads mRNA DIRECT Micro Kit | VERITAS | DB61021 |
| NEBNext Ultra Directional RNA Library Prep Kit for Illumina | New England Biolabs | E7645S |
| Dynabeads Oligo (dT)25 | VERITAS | DB61002 |
| AMPure XP beads | Beckman Coulter | A63881 |
| Bioanalyzer High Sensitivity DNA Analysis kit | Agilent | 5067-4626 |
| Bio-Mag Plus Concanavalin-A-coated beads | Polysciences | 86057-3 |
| **Software** | | |
| Cell Ranger (2.2.0) | https://support.10xgenomics.com/single-cell/software/downloads/latest | |
| Seurat package (4.1.0) | https://satijalab.org/seurat/ | |
| Signac (1.3.0) | https://stuartlab.org/signac/ | |
| R (4.0.3) | https://www.r-project.org/ | |
| Fiji | https://imagej.net/software/fiji/downloads | |
| Macs2 (2.2.7) | https://hbctraining.github.io/Intro-to-ChIPseq/lessons/05_peak_calling_macs.html | |
| Trimmomatic (0.39) | http://www.usadellab.org/cms/?page=trimmomatic | |
| STAR (2.6.1) | https://github.com/alexdobin/STAR | |
| RSEM (1.3.1) | https://github.com/deweylab/RSEM | |
| DESeq2 (1.28.0) | http://bioconductor.org/packages/deseq2/ | |
| Metascape (3.5) | http://metascape.org/ | |
| Bowtie2 (2.2.4) | http://bowtie-bio.sourceforge.net/bowtie2/index.shtml | |
| Homer (4.11) | http://homer.ucsd.edu/homer/ | |
| ChIPpeakAnno (3.11) | https://bioconductor.org/packages/release/bioc/html/ChIPpeakAnno.html | |
| SAMtools (0.1.19) | http://www.htslib.org/ | |
| **Other** | | |
| FACS Aria II | BD Biosciences | |
| NOVASeq6000 sequencer | Illumina | |

| Reagent/Resource | Reference or Source | Identifier or Catalog Number |
|---|---|---|
| Confocal laser microscope | Zeiss LSM 700 and LSM 800 | |
| Fluorescence microscope | Leica AF600 | |
| Illumina HiSEq 3000 | Illumina | |
| Illumina HiSEq-X | Illumina | |
| Tapestation | Agilent | |

## Mice

All mice used in this study were housed under a 12-h light/dark cycle with free access to food and water. All experiments were carried out according to the animal experimentation guidelines of Kyushu University, which comply with the National Institutes of Health Guide for the Care and Use of Laboratory Animals. Nestin-EGFP mice on a C57BL/6 background were kindly provided by M. Yamaguchi (Yamaguchi et al, 2000). To assess the proliferation of NSCs and neurogenesis, mice were injected intraperitoneally with EdU (50 mg/kg) once a day for a week. To generate conditional *Setd8* KO mice, which undergo CreERT2-mediated deletion of *Setd8* in adult NSCs in response to tamoxifen administration, *Setd8*$^{flox/flox}$ mice were crossed with *Nestin-CreERT2* mice and with a *Rosa26-YFP* reporter line (Oda et al, 2009; Imayoshi et al, 2006). *Nestin-CreERT2* mice carrying the *Rosa26-YFP* reporter transgene, but wild-type for *Setd8*, were used as controls. Tamoxifen (Sigma-Aldrich, T5648) was dissolved in sesame oil at 25 mg/mL. To induce elimination of *Setd8* in Nestin-expressing adult NS/PCs, 10 mg of tamoxifen was orally administered to 4- or 8-week-old (4w or 8w) *Setd8* cKO, het-cKO, and control mice with a feeding needle daily for four days. The mice were genotyped using two sets of PCR primers (*Setd8*$^{fl/fl}$: 5′-TGT AAG AGA ACT TTG AAT GG-3′ forward and 5′-AGG CAG GGG GAG GAT-3′ reverse primers; *Rosa26-YFP*: 5′-AAA GTC GCT CTG AGT TGT TAT-3′ 1st, 5′-GCG AAG AGT TTG TCC TCA ACC-3′ 2nd, and 5′-GGA GCG GGA GAA ATG GAT ATG-3′ 3rd primers).

## Tissue preparation

For tissue collection, mice were deeply anesthetized with an intraperitoneal injection of a mixture of 4 mg/kg midazolam, 0.3 mg/kg medetomidine, and 5 mg/kg butorphanol, followed by transcardial perfusion with 4% paraformaldehyde (PFA) in PBS. Brains were dissected and post-fixed overnight in the same fixative at 4 °C. For cryoprotection, fixed brains were transferred to 15% sucrose solution overnight at 4 °C, and then transferred to 30% sucrose solution overnight at 4 °C. The brains were embedded in optimal cutting temperature (OCT) compound (Tissue Tek, Sakura Finetek, 25608-930) and stored at −80 °C. Embedded frozen brains were serially sectioned in the coronal plane at 40 µm thickness and every sixth section was sequentially suspended in PBS in 6-well plates.

## Cell counting in brain sections

Cell counting was performed on every sixth coronal section (240 µm apart) containing DG at the same anatomical level. The total number of marker-positive cells per DG was estimated by multiplying the number counted in the series of sections by the inverse of the section sampling fraction. In brief, the number of counted cells was multiplied by 12 to obtain the total number of marker-positive cells in the bilateral DG per brain.

## Isolation of Nestin-EGFP+ cells from DG

Procedures for isolating Nestin-EGFP+ cells were modified from previously described dissociation and FACS protocols (Sakai et al, 2018). Briefly, P5, 12w, and 24w Nestin-EGFP mice were euthanized, and each brain was promptly harvested. The DG was quickly microdissected under a dissection scope and minced with a scalpel. The dissociated DG was transferred into prewarmed papain (25 U/mL) solution and incubated at 33 °C for 45 min. The suspension was then washed with 2 mL Minimum Essential Medium α (Gibco) containing 5% BSA to stop enzyme activity, mechanically dissociated by gentle pipetting with a fire-polished Pasteur pipette, and centrifuged at $130 \times g$ for 5 min. The cell pellet was then resuspended in 2 mL of Hanks' balanced salt solution (HBSS) containing 250 U/mL DNase and centrifuged at $130 \times g$ for 5 min, after which the cells were suspended in HBSS. The cells were sorted on a FACS Aria II (BD Biosciences), with 7-AAD (559925, BD Pharmingen) added to exclude nonviable cells from the analysis. Cells dissociated from the DG of WT mice were used to draw gates for EGFP+ and EGFP− cell populations. The EGFP+ cells were sorted into N2 medium and centrifuged at $200 \times g$ at 4 °C for 10 min. The supernatant was carefully removed, leaving 30 µL to resuspend the cell pellet. One microliter of cells was mixed with 10 µL Trypan Blue to determine cell viability. For scRNA-seq and scATAC-seq analyses, 700–1200 cells/µL were used. Libraries were prepared according to the manufacturer's instructions and sequenced using a NOVASeq6000 sequencer (Illumina).

## scRNA-seq analysis

Fastq files from sequencing were demultiplexed and then used as input for Cell Ranger (v2.2.0), and aligned to the mm10 reference genome for assignment of reads to single cells. To detect age-related gene expression changes, we conducted an integrated analysis using the IntegrateData function of the Seurat package (v4.1.0). Two datasets were confirmed as anchors using the FindIntegrationAnchors function, after which the anchors were used to integrate the three datasets using the IntegrateData function. We then performed a PCA analysis on the integrated dataset (Ximerakis et al, 2019). Using Seurat's non-linear dimensional reduction algorithm RunUMAP, we clustered cells using variable genes. Differential gene expression analysis was performed using the FindAllMarkers

function in Seurat. The known markers for NSCs and their progeny were obtained from previously published data (Hochgerner et al, 2018). We then subsetted the NSCs and their progeny from the analysis using Seurat's SubsetData function (subset.raw = T). After the first subsetting, re-clustering and PCA analysis were conducted to match clusters for DEGs analysis. To identify DEGs associated with aging in NSCs and their progeny, we used the FindMarker function in Seurat (test.use = MAST).

## scATAC-seq data pre-processing

Sequencing reads were aligned to the mm10 reference genome, and a cell-by-peak matrix was generated from fastq files using the software Cell Ranger (version 1.2.0) with default parameters. This returned barcoded fragment files which were loaded into Signac (1.3.0) (Stuart et al, 2019), an R (4.0.3) package, for downstream analysis using the standard Signac/Seurat pipeline. Macs2 (2.2.7) (Zhang et al, 2008) was run on the fragment files to call peaks using Signac's CallPeaks function. Fragments were mapped to the Macs2 called peaks and assigned to cells using Signac's FeatureMatrix function. Nucleosome signal strength and TSS enrichment for each cell were calculated using Signac's NucleosomeSignal and TSSEnrichment functions, respectively. Outliers in the QC metric categories were removed as per Signac's standard processing guidelines. After quality control, we obtained a total of 27,585 cells for further analysis, with a total of 203,829 peaks mapped to the nuclear genome and a median of 10,850 fragments per cell. Dimensionality reduction, clustering, and gene activity scores were determined using standard processes in Signac.

## Integration of scATAC-seq and scRNA-seq data as well as cell type annotation

We integrated our scATAC-seq and scRNA-seq from a previous study and annotated cell types and subtypes using the Seurat package. To improve computational efficiency and accuracy, we performed the integration in a stage-wise manner, matching developmental stages between studies. We performed canonical correlation analysis to generate a shared dimensionality reduction of the 'query' scATAC-seq gene activity calculated by Signac and the 'reference' scRNA-seq gene expression. We identified pairs of corresponding cells using highly variable genes in two datasets that anchor the two datasets together. To transfer cell type and subtype annotations from scRNA-seq to scATAC-seq cell populations, we used Seurat's label-transfer algorithm to leverage these anchors to predict cell types in scATAC-seq data. To investigate gene expression and chromatin accessibility in the same cell, we transferred the UMI counts from scRNA-seq to scATAC-seq using Seurat's label-transfer algorithm. To find epigenetic regulators that change in NSCs with age, a previously published gene list was employed.

## Immunohistochemistry

We performed immunohistochemistry as described previously (Sakai et al, 2018). Briefly, mouse brains were fixed in 4% PFA and 40-μm sections were cut with a cryostat. The sections were incubated with blocking solution (5% fetal bovine serum [FBS] and

0.3% Triton X-100) for 1 h at room temperature and then incubated with primary antibodies at 4 °C for one or two overnight periods. The following primary antibodies were used: chicken anti-Nestin (1:500, NES, Aves Labs); mouse anti-Ki67 (1:500, 550609, BD Biosciences); rat anti-Ki67 (1:500, 14-5698-84, eBioscience); chick anti-Gfap (1:500, AB5541, Millipore); goat anti-Sox2 (1:500, sc-17320, Santa Cruz Biotechnology); goat anti-DCX (1:500, sc-8066, Santa Cruz Biotechnology); rabbit anti-DCX (1:500, 4604, Cell Signaling Technology); mouse anti-NeuN (1:500, MAB-377, Millipore); rabbit anti-active Caspase 3 (1:500, AF-835, R&D Systems); chick anti-GFP (1:500, GFP-1010, Aves Labs); rat anti-GFP (1:500, Nacalai Tesque); anti-rabbit H4K20me1 (1:500, ab9051, abcam); and rat anti-BrdU (1:500, ab-6326, Abcam). After three washes in PBS, sections were incubated for 2 h with corresponding secondary antibodies: CF-488 donkey anti-mouse IgG (1:500, 20014, Biotium); CF-488 donkey anti-chicken IgG (1:500, 20166, Biotium); CF-555 donkey anti-rabbit IgG (1:500, 20038, Biotium); CF-555 donkey anti- mouse IgG (1:500, 20037, Biotium); CF-568 donkey anti-rat IgG (1:500, 20092, Biotium); CF-647 donkey anti-rabbit IgG (1:500, 20047, Biotium); and CF-647 donkey anti-goat IgG (1:500, 20048, Biotium). Nuclei were stained using Hoechst 33258 (Nacalai Tesque). Fluorescence images were obtained on a confocal laser microscope (LSM 700 and LSM 800, Zeiss). For immunostaining of cells, the cells were fixed with 4% PFA in PBS for 20 min and incubated for 2 h at room temperature with primary antibodies diluted in blocking solution, followed by incubation with the corresponding secondary antibodies for 1 h at room temperature. Images of stained cells were taken using a Leica AF600 fluorescence microscope. Quiescent (Ki67−) and active (Ki67+) NSCs were defined as cells showing radial glial-like morphology expressing GFAP and Sox2 or Nestin-EGFP.

## Image analysis

For analyzing the fluorescence intensities of H4K20me1 of NSCs in brain sections, ImageJ/Fiji was used. For brain sections, fluorescence signals of H4K20me1 were measured in quiescent NSCs and activated NSCs' nuclei as defined by their morphology and marker gene expression. These were then normalized to the background fluorescence of the tissue, i.e., fluorescence in the area where no cells are present.

## Cell culture

Mouse NSCs with multipotency and self-renewal capacity derived from the cortex at embryonic day 11 and passaged more than 20 times were used in this study. The NSCs were plated on poly-L-ornithine/fibronectin-coated dishes in Dulbecco's modified Eagle's medium (DMEM)/F-12 containing N2 supplement (7502048, Gibco) with 20 ng/mL basic fibroblast growth factor (bFGF; 100-18B, PeproTech), 20 ng/mL epidermal growth factor (EGF; AF-100-15, PeproTech), and 0.1 mg/mL penicillin/streptomycin/fungizone (HyClone) under 5% $CO_2$ at 37 °C. HEK293T human embryonic kidney cells were grown in DMEM (Nacalai Tesque) supplemented with 10% FBS. For Setd8 inhibition, we used UNC0379 (Selleck, S8572). The inhibitor was dissolved in DMSO and applied to the culture medium at a final concentration of 5 μM. Control cells were treated with an equivalent volume of DMSO.

## Knock-down experiments

To knock down *Setd8* expression, shRNA for *Setd8* was cloned into pLLX vector, which was generously provided by Drs. Z. Zhou and M. E. Greenberg. pLLX is a dual-promoter lentivirus vector constructed by inserting the U6 promoter-driven shRNA cassette 5′ to the ubiquitin-C promoter in the FUIGW plasmid (Lois, 2002; Zhou et al, 2006). We modified the pLLX vector to allow the expression of GFP together with a puromycin resistance gene under the ubiquitin-C promoter (Noguchi et al, 2016; Murao et al, 2019). shRNA for *Setd8*-expressing lentivirus constructs was generated by inserting the following oligonucleotides into the HpaI and XhoI sites of pLLX: *Setd8* KD_1-Forward, 5′-tGGGAGACTTTGTGGTAGAATAttcaagagaTATTCTACCACAAAGTCTCCCttttttggaac-3′, *Setd8* KD_1-Reverse, 5′-tcgagttccaaaaaaGGGAGACTTTGTGGTAGAATAtctcttgaaTATTCTACCACAAAG TCTCCCa-3′, *Setd8* KD_2-Forward, 5′-tGCCTCCTTC AAAGGACAAAGTttcaagagaACTTTGTCCTTT-GAAGGAGGCttttttggaac-3′, and *Setd8* KD_2-Reverse, 5′-tcgagttc-caaaaaaGCCTCCTTCAAAGGACAAAGTtctcttgaaACTTT GTCCTT TGAAGGAGGCa-3′. Lentiviruses were produced by transfecting HEK293T cells with the lentivirus constructs pCMV-VSV-G-RSV-Rev and pCAG-HIVgp using polyethylenimine. We used an shRNA containing a scrambled sequence as the control in our experiments.

## CRISPR-mediated knockout experiments

To knock out *Setd8* in cultured adult hippocampal NSCs, we utilized the lentiCRISPR v2 vector (Addgene #52961) for CRISPR/Cas9-mediated gene editing. Four different single-guide RNA (gRNA) sequences targeting *Setd8* were individually cloned into separate lentiCRISPR v2 plasmids. As a control, four distinct non-targeting gRNAs were also cloned into lentiCRISPR v2. The four *Setd8*-targeting constructs and the four control gRNA constructs were separately pooled (one tube per group). Lentiviral particles were produced and used to infect adult hippocampal NSCs. The gRNA sequences were selected based on validated sequences from the Mouse CRISPR Knockout Pooled Library (Brie, Addgene #73633). The specific sequences used were as follows: Setd8_gRNA1: GAACTCGGTTGCCCATCATG, Setd8_gRNA2: CTGCTTACCCCGTCGCTGCG, Setd8_gRNA3: TCACAGATTTC TACCCTGTG, Setd8_gRNA4: AGCCCTGAAAAAGTCCCTCA; Ctrl_gRNA1: AAAAAGTCCGCGATTACGTC, Ctrl_gRNA2: AACGGTAGCGTACCCGTGAA, Ctrl_gRNA3: CCATCGGTTCGA CTTACCGC, Ctrl_gRNA4: TAAGACTGGGTGTCCCGCGT.

## Real-time qPCR analysis

Total RNA was isolated from cells using Sepasol-RNA I Super G (Nacalai Tesque) following the manufacturer's instructions. RNA quality of all samples was checked by spectrophotometer. According to the kit protocol, reverse transcription reactions were carried out using a SuperScript VILO cDNA Synthesis Kit (Life Technologies). qRT-PCR was performed with SYBR green fluorescent dye using Step One Plus (Applied Biosystems) and Mx3000 (Stratagene). β-actin was used as an endogenous control to normalize samples.

## Ethynyl deoxyuridine (EdU) labeling and detection

For EdU experiments, 10 mM EdU was added to the culture medium, and the cells were fixed 1 h later. EdU staining was performed using a Click-iT EdU Alexa Fluor 555 or 647 Imaging Kit (Life Technologies) according to the supplier's protocol. Images of stained cells were taken using a Leica AF600 fluorescence microscope.

## Behavioral test

The behavior of female mice is affected by the estrous cycle. Therefore, we used male mice to exclude the effects of the estrous cycle in this study and focus on the effects of impaired NSC function. NPRT was conducted as previously described (Manning et al, 2012; Mumby et al, 2002). In brief, mice were habituated to a white plastic chamber ($50 \times 50 \times 30$ [H] cm) for 10 min on each of the three days prior to the NPRT. After habituation, for a familiarization session, each mouse was left in the testing chamber, in which two identical objects were placed in adjacent corners, and allowed to freely explore the objects for 5 min and then taken back to its home cage for 5 min. Before a testing session, one object was shifted to a new location near the corners in the chamber. In a testing session, the mouse was again placed in the testing chamber and allowed to explore the familiar and displaced objects for 5 min. All sessions were recorded with an overhead video tracking system, and exploration behavior was defined as activities such as whisking, sniffing, and rearing against the object. Time spent exploring each object during the familiarization and testing sessions was automatically scored. Exploration ratios to each object were calculated using the equation t novel or t familiar/(t novel + t familiar), as described previously (Manning et al, 2012). The testing chamber and objects were washed with 70% ethanol before the next mouse was tested. Contextual fear conditioning test was conducted as previously described (Doi et al, 2021). In brief, on the conditioning trial (Day 1), each mouse was placed in a conditioning chamber ($17 \times 10 \times 10$ cm, 100 lx) with clear acrylic walls and a shock-grid floor made of stainless steel rods (2 mm in diameter, spaced 5 mm apart) and allowed to explore freely for 1 min. A 65-dB tone was then presented for 30 s. During the final 2 s of tone presentation, a 0.5-mA foot shock was delivered. The set of tone and foot shock was repeated 3 times automatically using a tone and shock generator controlled by Time FZ software (O'Hara & Co.). The next day (Day 2), to evaluate contextual fear memory, mice were placed in the same context arena without tone or shock for 5 min. Overhead infrared cameras recorded behavior, and the duration of freezing (i.e., absence of movement except for breathing) was automatically calculated using Image FZ2 software.

## mRNA isolation and cDNA library preparation for bulk RNA-seq

We used 50,000 cultured NSCs to construct a library for each sample for RNA-seq ($n = 3$). mRNAs were collected using a Dynabeads mRNA DIRECT Micro Kit (Invitrogen), and the purified mRNAs were subjected to cDNA library construction using a NEBNext Ultra Directional RNA Library Prep Kit for Illumina (New England Biolabs) following the manufacturer's protocols. Briefly, polyadenylated mRNA was isolated directly from the cells using Dynabeads Oligo (dT)25 (Invitrogen). mRNA was fragmented in NEBNext First Strand Synthesis Reaction Buffer by heating at 94 °C for 15 min. First-strand cDNA was reverse-transcribed from the fragmented mRNA and then used as template

to synthesize second-strand cDNA. dTTP was replaced with dUTP during the second-strand cDNA synthesis. The cDNA was end-repaired, dA-tailed, and ligated with NEBNext Adaptor. The second-strand cDNA containing dUTP was digested with USER enzyme, and a sequencing tag and barcode were introduced through a 12–15-cycle PCR amplification. The cDNA library was purified using AMPure XP beads (Beckman Coulter). The quality of the cDNA library was assessed using the Bioanalyzer High Sensitivity DNA Analysis kit (Agilent).

## Bulk RNA-seq and bioinformatics analysis

RNA-seq was performed with 50-bp single-end sequencing using an Illumina HiSEq 3000, described previously (Matsuda et al, 2019) with minor modifications. Briefly, obtained reads were processed with the Trimmomatic tool kit (Bolger et al, 2014) to remove short (<30 bp) and low-quality (quality score < 20) reads, followed by trimming of the adaptor sequence. Processed reads were aligned to the mouse reference genome (mm10) using STAR (Dobin et al, 2013). The DESeq2 package (Love et al, 2014) was used for differential gene expression analysis. Gene expression levels were quantified as transcripts per million. GO analysis was performed using Metascape. We analyzed GO term enrichment in the Biological Process and Mouse Phenotype categories. GSEA was carried out using signal-to-noise as the ranking metric and with the 'weighted' scoring scheme.

## Bulk ATAC-seq

We used 50,000 cells and the same conditions as for RNA-seq to construct a library for each sample for ATAC-seq ($n = 2$ for each group). ATAC-seq was performed as previously described (Buenrostro et al, 2015) with minor modifications. Briefly, samples retrieved from dishes were pelleted and incubated with cold lysis buffer (10 mM Tris·HCl [pH 7.4], 10 mM NaCl, 3 mM MgCl$_2$, and 0.1% IGEPAL CA-630). After removing the lysis buffer by centrifugation, samples were immediately subjected to a transposition reaction at 37 °C for 30 min with 2.5 µL transposase (Illumina Nextera DNA Preparation Kit). Transposed DNA was purified using a FastGene Gel/PCR Extraction Kit (Nippon Genetics). The purified DNA was amplified for 12 to 15 cycles using adapters from the Nextera index kit (Illumina) according to the manufacturer's instructions, and PCR purification was performed using AMPure XP beads (Beckman Coulter) to remove large fragments and remaining primers. Quality of the cDNA library was assessed by Tapestation using an HS DNA Kit (Agilent). An Illumina HiSEq-X was used to perform 150-bp paired-end sequencing. Sequencing reads were mapped to the mouse reference genome (mm10) from the University of California, Santa Cruz (UCSC) Genome Browser using Bowtie2 (v2.2.4) (Langmead and Salzberg, 2012). Open chromatin regions were analyzed using MACS2 software (v2.1.0.20141030) against BAM data compiled by BEDTools. We calculated RPKM of the region of peaks merged with control and KD using bedtools-multicov, and define DARs as regions with fold change in RPKM ≥ 1.5. The consensus sequence was identified by importing whole sequences of the open or closed DARs using homer. We found DARs-annotated genes using ChIPpeakAnno (Zhu et al, 2010).

## CUT&RUN against H4K20me1

We followed CUT&RUN described in Skene and Benioff (Skene and Henikoff, 2017), with minor variations. Samples were pelleted and resuspended in wash buffer. Bio-Mag Plus Concanavalin-A-coated beads (86057-3, Polysciences) were added to cells (diluted in binding buffer) to bind nuclei to beads. The samples were rotated for 10 min at room temperature and the supernatant was discarded using a magnetic stand. Antibody (150 µL) (anti-H4K20me1) containing digitonin was added and the samples were rotated at 4 °C overnight. The supernatant was discarded using the magnetic stand. Beads were collected and washed 3× with digitonin-containing wash buffer before adding pA-MNase to the beads. After a 1-h incubation with AG-MNase at 4 °C, the beads were washed 3× and resuspended in digitonin-containing wash buffer. After equilibration on ice, 100 mM CaCl$_2$ was added and the samples were incubated for 30 min with agitation from 15 min. The reaction was stopped by adding stop buffer and the samples were incubated at 37 °C for 30 min to release chromatin. The supernatant was retrained using the magnetic stand. DNA was isolated by protein degradation with 10% SDS and Proteinase K and phenol–chloroform–isoamyl alcohol extraction to maximally retain chromatin. Chromatin was dissolved in low-EDTA TE buffer and analyzed by TapeStation using the HS DNA Kit (Agilent).

## ChIP-seq and data analysis

Deep sequencing of immunoprecipitated chromatin was performed using the Illumina HiSEq-X system, as described previously (Matsuda et al, 2019). Briefly, sequencing libraries were made using the NEBNext ChIP-Seq Library Prep Master Mix Set for Illumina and NEBNext Multiplex Oligos for Illumina (New England Biolabs). Paired-end 150-bp sequence reads were trimmed based on read length and read quality using the Trimmomatic tool kit and aligned to the mouse genome (mm10) using Bowtie2 (v2.2.4) (Langmead and Salzberg, 2012). All redundant reads were removed from further analysis. SAM files were converted to the BAM format using SAMtools (v0.1.19) (Li et al, 2009). We calculated RPKM in the gene body of all genes for control and KD using bedtools-multicov and defined genes with decreased H4K20me1 in KD as those having H4K20me1 RPKM values < 0.5× times control in the gene body. For boxplot analysis, the mouse genome was partitioned into 500-bp bins, and signal enrichment was calculated as RPKM mapped reads.

## Adult hippocampal NSC establishment

In this study, we used adult mouse hippocampal NSCs that retained multipotency and self-renewal capacity and had been passaged more than ten times. We established adult NSCs using the same method that we previously reported (Murao et al, 2024). Briefly, the cell preparation process was as follows. Three 8-week-old C57BL/6 mice were euthanized, and their brains were immediately harvested. The DG was quickly microdissected under a dissection microscope, minced with a scalpel, and transferred into a prewarmed papain solution (25 U/mL). The tissue was incubated at 37 °C for 30 min, after which enzymatic activity was halted by washing the suspension with 2 mL of Minimum Essential Medium α (Gibco, 12571063) containing 5% BSA. The tissue was then mechanically dissociated by gentle pipetting with a fire-polished Pasteur pipette and centrifuged at $130 \times g$ for 5 min. The resulting cell pellet was resuspended in 2 mL of Hanks' Balanced Salt Solution (HBSS)

containing 250 U/mL DNase and centrifuged again at $130 \times g$ for 5 min. After the final centrifugation, the cells were suspended in HBSS and plated onto poly-l-ornithine/laminin-coated dishes. Cells were maintained in DMEM/F12 supplemented with N2 (Gibco, 17502048), 20 ng/mL basic fibroblast growth factor (bFGF; PeproTech, 100-18B), 20 ng/mL epidermal growth factor (EGF; PeproTech, AF-100-15), a 1:1000 dilution of B27 supplement (Gibco, 17504-044), and 0.1 mg/mL penicillin/streptomycin/fungizone (HyClone, SV30079.01), referred to as proliferation medium. For Setd8 inhibition, we used UNC0379 (Selleck, S8572). The inhibitor was dissolved in DMSO and applied to the culture medium at a final concentration of 5 μM. Control cells were treated with an equivalent volume of DMSO.

### Quantification and statistical analysis

No statistical methods were used to pre-determine sample sizes. The statistical analyses were done afterward without interim data analysis. No data points were excluded. We did not use any methods to determine whether the data met assumptions of the statistical approach. All data were collected and processed randomly. Unpaired Student's t tests and the Wilcoxon rank-sum test were used to calculate the $p$ value for pairwise comparisons. For multiple comparisons, $p$ values were calculated using one-way ANOVA with the Tukey post hoc test and Wilcoxon rank-sum test. The significance of the enrichment of transcription factors in indicated regions or sites was determined using the hypergeometric distribution. The exact values of $n$ (sample size) are provided in the figures and figure legends. We considered a $p$ value $< 0.05$ to be statistically significant. Data represent mean + SEM.

## Data availability

The scRNA-seq, scATAC-seq, RNA-Seq, ATAC-seq, and ChIP-seq data obtained in this study have been uploaded to NCBI GEO datasets, under accession number GSE256417.

The source data of this paper are collected in the following database record: biostudies:S-SCDT-10_1038-S44318-025-00455-8.

## Peer review information

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

## Acknowledgements

We thank H. Nakashima, S. Katada, H. Noguchi, and E. Hatai for discussions; Y. Nakagawa and R. Nakayama for excellent secretarial assistance; I. Smith for proofreading the manuscript; I. Imayoshi for sharing *Nestin-CreERT2* mice; Y. Nakabeppu and D. Reinberg for sharing *Setd8*fl/fl mice; and F. Costantini for sharing *Rosa26-YFP* reporter mice. We appreciate technical assistance from the Research Support Center, Research Center for Human Disease Modeling, Kyushu University Graduate School of Medical Sciences, which is partially supported by the Mitsuaki Shiraishi Fund for Basic Medical Research. The LiSCO at the Nara Institute of Science and Technology (NAIST) was instrumental in this study. This work was supported by JSPS KAKENHI Grant Numbers JP16H06527 (to KN), JP18K14820, JP21H02808, JP23K18451, JP16H06279 (PAGS), (all to TM), and JP20J20975 (to SM), and by the Nakajima Foundation and the Mochida Foundation (to TM) as well as the JST FOREST Program, Grant Number JPMJFR231Z (to TM). This study was supported by AMED through grant numbers JP20gm1310008 and JP24wm0625306 to K.N.

## Author contributions

**Shuzo Matsubara**: Conceptualization; Funding acquisition; Investigation; Visualization; Writing—original draft; Writing—review and editing. **Kanae Matsuda-Ito**: Validation; Investigation; Visualization. **Haruka Sekiryu**: Investigation. **Hiroyoshi Doi**: Investigation. **Takumi Nakagawa**: Investigation. **Naoya Murao**: Investigation. **Hisanobu Oda**: Conceptualization; Methodology. **Kinichi Nakashima**: Conceptualization; Supervision; Funding acquisition. **Taito Matsuda**: Conceptualization; Supervision; Funding acquisition; Validation; Investigation; Visualization; Writing—original draft; Project administration; Writing—review and editing.

Source data underlying figure panels in this paper may have individual authorship assigned. Where available, figure panel/source data authorship is listed in the following database record: biostudies:S-SCDT-10_1038-S44318-025-00455-8.

## Disclosure and competing interests statement

The authors declare no competing interests.

