## [Peer Review File · The EMBO Journal]

Epigenetic Regulation of Neural Stem Cell Aging in the Mouse Hippocampus by Setd8 Downregulation

Shuzo Matsubara, Kanae Matsuda-Ito, Haruka Sekiryu, Hiroyoshi Doi, Takumi Nakagawa, Naoya Murao, Hisanobu Oda, Kinichi Nakashima, and Taito Matsuda

Corresponding authors: Taito Matsuda (matsuda.taito@naist.ac.jp) , Kinichi Nakashima (nakashima.kinichi.718@m.kyushu-u.ac.jp)

Review Timeline:

Submission Date:	16th Jun 24
Editorial Decision:	17th Aug 24
Revision Received:	28th Feb 25
Editorial Decision:	14th Mar 25
Revision Received:	5th Apr 25
Accepted:	16th Apr 25

Editor: Daniel Klimmeck

Transaction Report:

Dear Dr Matsuda,

Thank you again for the submission of your manuscript (EMBOJ-2024-118177) to The EMBO Journal. Please accept my sincere apologies for getting back to you with this unusual delay due to protracted referee input and detailed discussion in the editorial team. As mentioned earlier, your study was sent out to three reviewers with expertise developmental neuroscience and epigenetics, however one expert got delayed and was in the end not able to send comments. The reports of the other two referees are enclosed below.

As you will see from the experts' reports, the referees acknowledge the analysis and potential interest of your results. However, they also express major concerns regarding completeness and robustness of the findings, which need to be addressed thoroughly to make them supportive of publication in the EMBO Journal. The reviewers also raise a number of issues related to the data presentation, additional controls and improved methods annotation required, statistics applied and overall discussion of related literature, that would need to be conclusively addressed to achieve the level of robustness and clarity needed for The EMBO Journal.

Given the overall interest stated and broader angle of your findings, we are able to invite you to revise your manuscript experimentally to address the referees' comments. I need to stress though that we do require strong support from the referees on a revised version of the study in order to move on to publication of the work.

In light of the extensive experimentation requested, I would appreciate if you could contact me during the next weeks for exchange e.g. a video call to discuss your perspective on the comments and potential plan for revisions.

Please feel free to contact me if you have any questions or need further input on the referee comments.

When submitting your revised manuscript, please carefully review the instructions below.

Please feel free to approach me any time should you have additional questions related to this.

Thank you for the opportunity to consider your work for publication.

I look forward to your revision.

Kind regards,

Daniel Klimmeck

Daniel Klimmeck, PhD
Senior Editor
The EMBO Journal

Instruction for the preparation of your revised manuscript:

- 1) a .docx formatted version of the manuscript text (including legends for main figures, EV figures and tables). Please make sure that the changes are highlighted to be clearly visible.
- 2) individual production quality figure files as .eps, .tif, .jpg (one file per figure).
- 3) a .docx formatted letter INCLUDING the reviewers' reports and your detailed point-by-point response to their comments. As part of the EMBO Press transparent editorial process, the point-by-point response is part of the Review Process File (RPF), which will be published alongside your paper.

4) a complete author checklist, which you can download from our author guidelines ([https://wol-prod-cdn.literatumonline.com/pb-assets/embo-site/Author Checklist%20-%20EMBO%20J-1561436015657.xlsx](https://wol-prod-cdn.literatumonline.com/pb-assets/embo-site/Author%20Checklist%20-%20EMBO%20J-1561436015657.xlsx)). Please insert information in the checklist that is also reflected in the manuscript. The completed author checklist will also be part of the RPF.

6) It is mandatory to include a 'Data Availability' section after the Materials and Methods. Before submitting your revision, primary datasets produced in this study need to be deposited in an appropriate public database, and the accession numbers and database listed under 'Data Availability'. Please remember to provide a reviewer password if the datasets are not yet public (see <https://www.embopress.org/page/journal/14602075/authorguide#datadeposition>).

7) Our journal encourages inclusion of *data citations in the reference list* to directly cite datasets that were re-used and obtained from public databases. Data citations in the article text are distinct from normal bibliographical citations and should directly link to the database records from which the data can be accessed. In the main text, data citations are formatted as follows: "Data ref: Smith et al, 2001" or "Data ref: NCBI Sequence Read Archive PRJNA342805, 2017". In the Reference list, data citations must be labeled with "[DATASET]". A data reference must provide the database name, accession number/identifiers and a resolvable link to the landing page from which the data can be accessed at the end of the reference. Further instructions are available at .

8) At EMBO Press we ask authors to provide source data for the main and EV figures. Our source data coordinator will contact you to discuss which figure panels we would need source data for and will also provide you with helpful tips on how to upload and organize the files.

Numerical data can be provided as individual .xls or .csv files (including a tab describing the data). For 'blots' or microscopy, uncropped images should be submitted (using a zip archive or a single pdf per main figure if multiple images need to be supplied for one panel). Additional information on source data and instruction on how to label the files are available at .

9) We replaced Supplementary Information with Expanded View (EV) Figures and Tables that are collapsible/expandable online (see examples in <https://www.embopress.org/doi/10.15252/emboj.201695874>). A maximum of 5 EV Figures can be typeset. EV Figures should be cited as 'Figure EV1, Figure EV2' etc. in the text and their respective legends should be included in the main text after the legends of regular figures.

11) For data quantification: please specify the name of the statistical test used to generate error bars and P values, the number (n) of independent experiments (specify technical or biological replicates) underlying each data point and the test used to calculate p-values in each figure legend. The figure legends should contain a basic description of n, P and the test applied. Graphs must include a description of the bars and the error bars (s.d., s.e.m.).

Please remember: Digital image enhancement is acceptable practice, as long as it accurately represents the original data and conforms to community standards. If a figure has been subjected to significant electronic manipulation, this must be noted in the figure legend or in the 'Materials and Methods' section. The editors reserve the right to request original versions of figures and

the original images that were used to assemble the figure.

The revision must be submitted online within 90 days; please click on the link below to submit the revision online before 15th Nov 2024.

Referee #1:

Adult neurogenesis in the rodent brain is known to decline during chronological aging, but the underlying molecular mechanisms are still poorly understood. The manuscript by Matsubara et al investigates the transcriptional and epigenetic changes that take place during aging of hippocampal neural stem cells (NSCs), and some of their progeny, from neonatal to mature adult stages. To this end, the authors have combined a genetic tracing strategy (using Nestin-EGFP mice) with a single-cell omics approach. They characterized age-related changes in the transcriptome (scRNA-seq) and chromatin accessibility (scATAC-seq) from P5 to 24 Weeks, highlighting pathways and putative transcription factors with activity impaired. Interestingly, alterations in chromatin landscape are associated and precede changes in gene expression. Moreover, age-related changes in the epigenome and transcriptome are not restricted to NSCs, being also observed in their neuronal committed progeny.

Focusing on potential epigenetic regulators with altered gene expression during aging, the authors found Setd8 to be strongly downregulated in NSCs, with the levels of the histone modification it catalyzes (H4K20me1) decreasing in both quiescent and active NSCs. Importantly, conditional (acute) knock-out of Setd8 specifically in NSCs results in a very significant reduction of the number of this cell population (both active and quiescent NSCs) in young adult stages. This was also associated with decreased neurogenesis and reduced performance in a hippocampal memory function test, altogether suggesting that knocking out Setd8 mimics premature aging.

To characterize the molecular changes in NSCs upon reduced Setd8 expression, the authors used an shRNA-mediated knock-down approach in cultured NSCs. The impact of reduced Setd8 expression was assessed by transcriptional, H4K20me1 and ATAC-seq profiling. An important conclusion was that down-regulation of Setd8 mimics age-dependent changes in the expression of a vast number of genes (including up-regulation of quiescence and senescence-related genes) and that a significant fraction of age-related changes results from the direct action of Setd8 in modifying the chromatin landscape. Overall, this is a scientifically sound study which identifies Setd8 as an important mediator of epigenetic changes (and associated alterations in gene expression) undergone by NSCs during chronological aging. Decreased neurogenesis (and a reduced number of hippocampal NSCs) is a well-known hallmark of aging of the rodent brain. A few prior studies have applied single-cell transcriptomics to the aging hippocampal niche, describing differences in cellular dynamics during aging. However, the identification of genes deregulated in NSCs during this process (and the associated epigenetic drivers) remains poorly characterized. This study is therefore an important contribution to the field.

Overall, the experimental work is robust and of good quality. However, a major concern relates to the quality of experiments knocking down Setd8 expression in NSC cultures: i) NSCs of embryonic (cortex) origin were used, and ii) control lentivirus as I understand does not express any scramble (or similar) shControl. I suggest the authors validate some of their findings (impact on proliferation, changes in expression of selected genes), using a more suitable cellular model and knockdown approach (e.g. Setd8 cKO in hippocampal NSCs).

Additional points:

Figure 2E: 12W sample looks very different from 24w in MN1 and MN2, although heatmap should display regions shown in Figure 1D.

Page 12 (line 247): please clarify which group of 171 genes is referred here.

Figure 3C: I suggest to include in the comparison the levels of H4K20me1 at P5, given the differences in transcript levels shown in Figure 3B.

Page 14 (line 295): why is LaminB1 expression invoked here, please explain better.

Figure 5G: can the authors better clarify whether genes that are deregulated upon acute Setd8 KD and overlap with genes that change in NSCs with aging, change expression in the same direction? This is an important point if Setd8 mediates aging-related changes.

Figure 5 (I-N and P): graphs show regulation of individual genes from the genomics data. These should be individually validated by qPCR using a better model and approach (related to my main point above).

Figure 5 (O-P): can the authors provide a better description of the experiment in figure legend (long-term suppression of Setd8 at 7 days post-infection).

Figure 2F: typo "Gained-pen"

Referee #2:

The manuscript by Matsuda and colleagues investigates the possible role of epigenetic regulation, specifically monomethylation of H4K20, in the process of aging of the Neural Stem Cells within the hippocampus. This is interesting for the involvement of this kind of cells in sustaining adult neurogenesis, downstream to this, the processing of memory.

The authors performed scRNA+ATACseq of nestin-EGFP+ cells from P5, 12W and 24W hippocampi. Among the genes which changed expression in NSCs and their progeny during aging they found molecular signature of cellular quiescence and senescence. Since they found a strong chromatin accessibility alteration (mainly closure) in relation with age, they looked at epigenetic-related players among the deregulated gens, finding Setd8 (the sole enzyme mediating H4K20me1) as the most deregulated (decrease with age). Consistently, H4K20me1 levels decreased with age in NSCs.

Making the mouse KO for Setd8 specifically in NSCs, the authors found that this lack led to decrease in NSC number (with those activated really low in %) and, as consequence, low neurogenesis and impairment in memory. The lack of Setd8 drive to an increase of senescence features in NSCs with can be mapped transcriptomically and with the lens of chromatin accessibility assay. Thus, this manuscript describes a mechanism for the early decline of NSC (downregulation of Setd8, less H4K20me1, less chromatin accessibility, decrease in gene transcription related) which leads to decrease in neurogenesis and physiological properties (memory) which precede chronological aging.

While the story is interesting and of worth there are missing points and limitations that at this stage dampen the enthusiasm for the manuscript in this form.

Major points:

- Can the authors clarify by which event(s) the NSCs are less in the Setd8 mutant? Data seem ruled out differentiation. Is it for Caspase-independent cell death? Is it for stop the doublings (the time window seems too short for this but approaches like the BrdU labelling during the TAM administration may help).
- Other experimental settings for assessing hippocampal-related memory should be conducted. Males and female should be grouped separately.
- Genomic localization of Setd8 should be mapped and related with both H4K20me1 and ATAC.
- What is the link between H4K20me1 levels in the gene body and the chromatin accessibility differences (mainly closure) which are evident upon physiological aging of NSC or Setd8 KO/KD?
- An interesting question would be: are the changes caused by Setd8 repression (linked with senescence) reversible if Setd8 comes back to normal levels? This would imply that the NSC aging in DG may be slow down (or reverted) with Setd8 supplementation. Can the authors address this question experimentally?

Minor Points:

- Since it is true that the number of NSCs at 10W in mutants is comparable of the number at 30W in WT animals I would not state that the biological aging is accelerated (Pag 14). In fact the profiling of the aging -associated molecular and phenotypic signatures is missing there
- Fig.5G and H: it is not entirely clear to me what is this analysis. Could you please clarify?
- Fig.5: Describe better in the figure the timing of the experiments (improve the labelling). As it is, the figure is misleading for the reader.
- Fig. 6D: not entirely clear what group of genes are "with H4K20me1 modification in their gene body and with RPKM values less than $< 0.5 \times$ those of Ctrl in the Setd8 KD NSCs"
- Please use the correct international nomenclature for genes.

Additional suggestions:

- scRNA+ATAC should be done in the Setd8 KO too.

Dear Dr. Klimmeck,

We enclose the revised version of our manuscript entitled “**Epigenetic Regulation of Neural Stem Cell Aging in the Mouse Hippocampus by Setd8 Downregulation**” by Matsubara *et al.* (EMBOJ-2024-118177), which we submitted for publication as an Article in *The EMBO Journal*.

We thank both reviewers for their constructive suggestions for improving our manuscript. We would also like to sincerely thank you for granting us an extension on the revision deadline in light of my transition to a new institution.

Our point-by-point responses to comments provided by each reviewer are as follows:

<Responses to the comments by Reviewer #1>

Comment 1-0

Overall, the experimental work is robust and of good quality.

Response 1-0

We thank you for this positive comment. Before conducting the revision experiments, we engaged in careful discussions with the editor to ensure that our approach to those experiments was appropriate. We hope that our additional experiments have successfully addressed the points you raised.

A major concern:

Comment 1-1

i) NSCs of embryonic (cortex) origin were used, and ii) control lentivirus as I understand does not express any scramble (or similar) shControl.

I suggest the authors validate some of their findings (impact on proliferation, changes in expression of selected genes) using a more suitable cellular model and knockdown

approach (e.g., *Setd8* cKO in hippocampal NSCs).

Response 1-1

We thank you for this valuable comment. We would like to clarify that we used an shRNA containing a scrambled sequence as the control in our experiments. To ensure this point is clear, we have revised the Methods section to explicitly state the use of a scrambled shRNA as the control (page 33, lines 751–752).

To further address your comment, we investigated the effects of *Setd8* knockdown using a lentiviral *Setd8* knockdown vector, as well as *Setd8* deletion using Cas9, on the proliferation of adult hippocampal NSCs. After infection, cells were subjected to puromycin selection to enrich successfully transduced cells. We found that both knockdown and knockout of *Setd8* reduced proliferation and altered *Notch2* and *Lmnbl1* expression in adult NSCs (as shown below, Fig. S4D–H and Fig. S7D–F), as observed when using embryonic NSCs with repeated passages.

(D) Experimental scheme for assessing the effect of *Setd8* knockdown (KD) (E and F) or knockout (KO) (G and H) on proliferation of adult hippocampal NSCs *in vitro*.

(E) Representative images of staining for GFP (green), Sox2 (cyan), EdU (red), and Hoechst (gray) at 3 days after Ctrl or *Setd8* KD viral infection. Scale bar, 50 μ m.

(F) Proportion of GFP+ EdU+ cells among GFP+ cells (E) at 3 days after Ctrl or *Setd8* KD viral infection (n = 4 each). * $p < 0.01$ by Student's t test.

(G) Representative images of staining for Sox2 (cyan), EdU (red), and Hoechst (gray) at 3 days after infection with a vector containing both Cas9 and *Setd8* gRNA, or a control vector, followed by puromycin selection.

(H) Proportion of EdU+ Sox2+ cells among Sox2+ cells (G) at 3 days after Cas9 and gRNA viral infection (n = 4 each). * $p < 0.01$ by Student's t test.

(D) Experimental scheme for assessing the effect of *Setd8* KD (E) or KO (F) on proliferation of adult hippocampal NSCs *in vitro*.

(E,F) qRT-PCR analysis of *Notch2* and *Lmnbl1* mRNA levels in *Setd8* KD (E) or KO (F) NSCs (n = 4). * $p < 0.05$ by Student's t test.

We have added the following sentences in the main text (page 13, lines 274–279) and Methods (page 33, lines 754–768) of our revised manuscript:

Main text:

“We also investigated the effects of *Setd8* KD using a lentiviral *Setd8* KD vector, as well

as *Setd8* deletion using Cas9, on the proliferation of adult hippocampal NSCs. After infection, cells were subjected to puromycin selection to enrich successfully transduced cells. We found that both knockdown and knockout of *Setd8* reduced the proliferation of adult hippocampal NSCs (Figures S4D–H).”

Methods:

“CRISPR-mediated knockout experiments

To knockout *Setd8* in cultured adult hippocampal NSCs, we utilized the lentiCRISPR v2 vector (Addgene #52961) for CRISPR/Cas9-mediated gene editing. Four different single-guide RNA (gRNA) sequences targeting *Setd8* were individually cloned into separate lentiCRISPR v2 plasmids. As a control, four distinct non-targeting gRNAs were also cloned into lentiCRISPR v2. The four *Setd8*-targeting constructs and the four control gRNA constructs were separately pooled (one tube per group). Lentiviral particles were produced and used to infect adult hippocampal NSCs. The gRNA sequences were selected based on validated sequences from the Mouse CRISPR Knockout Pooled Library (Brie, Addgene #73633). The specific sequences used were as follows: *Setd8*_gRNA1: GAACTCGGTTGCCCATCATG, *Setd8*_gRNA2: CTGCTTACCCCGTCGCTGCG, *Setd8*_gRNA3: TCACAGATTTCTACCCTGTG, *Setd8*_gRNA4: AGCCCTGAAAAAGTCCCTCA, *Ctrl*_gRNA1: AAAAAGTCCGCGATTACGTC, *Ctrl*_gRNA2: AACGGTAGCGTACCCGTGAA, *Ctrl*_gRNA3: CCATCGGTTGCGACTTACCGC, *Ctrl*_gRNA4: TAAGACTGGGTGTCCCGCGT.”

Additional points:

Comment 1-2

Figure 2E: 12W sample looks very different from 24w in MN1 and MN2, although heatmap should display regions shown in Figure 1D.

Response 1-2

Laboratory of Neural Regeneration and Brain Repair

Division of Biological Science
Graduate School of Science and Technology,
Nara Institute of Science and Technology (NAIST)
8916-5, Takayama-cho, Ikoma, Nara, 630-0192, Japan
Tel +81-743-72-5556

To THE *EMBO JOURNAL* -5-

We apologize for the confusing explanation in our manuscript. The heatmaps in Figure 2E show the chromatin accessibility of the regions indicated in Figure 2D. Changes in chromatin accessibility near the up- and downregulated genes identified in Figure 1 are shown in Figure 2C using a boxplot. We have amended the main text and figure legend to explain these points clearly in the revised manuscript as follows:

Main text (page 10, lines 216–page 11, 225):

“We next assessed chromatin accessibility near the up- and downregulated DEGs in clusters of NSCs and their progeny with age and observed increased and decreased accessibility, respectively (Figure 2C).”

“To identify regions with altered chromatin accessibility across the whole genome, rather than at individual gene loci, we detected the differentially accessible regions (DARs) in the identified clusters with age using MACS call and defined gained-open or -closed DARs ($p < 0.05$ and fold change ≥ 1.5) at both 12w and 24w compared to P5. In NSCs, 751 DARs were found to have gained-open status, while 7,488 had gained-closed status (Figures 2D and 2E).”

Figure 2 legend (page 58 lines 1344-1351; page 58, line1355) :

“(C) Box plots showing the chromatin accessibility (gene activity score) around up- and downregulated DEG loci indicated in Figure 1 (E-H) (≥ 1.25 -fold, up in NSC cluster (n = 244), up in NPC cluster (n = 184), up in IMN1 cluster (n = 159), up in IMN2 cluster (n = 129), down in NSC cluster (n = 184), down in NPC cluster (n = 84), down in IMN1 cluster (n = 19), down in IMN2 cluster (n = 26)).

(E) Heatmaps showing the chromatin accessibility of the regions indicated in (D). “

Comment 1-3

Page 12 (line 247): please clarify which group of 171 genes is referred here.

Response 1-3

Laboratory of Neural Regeneration and Brain Repair

Division of Biological Science
Graduate School of Science and Technology,
Nara Institute of Science and Technology (NAIST)
8916-5, Takayama-cho, Ikoma, Nara, 630-0192, Japan
Tel +81-743-72-5556

To THE *EMBO JOURNAL* -6-

This group of 171 genes represents a list of epigenetics-related genes previously defined in Matsuda *et al.*, *Neuron* (2019). These genes were identified and categorized in that study as being involved in epigenetic regulation. We have now described this point in the manuscript to ensure clarity for the readers, as follows:

“By overlapping a previously defined list of epigenetics-related genes⁴⁵ (n = 171) with the list of upregulated or downregulated DEGs in NSCs (Table S3)”

Additionally, the complete list of the 171 epigenetics-related genes is provided as Supplementary Table 3.

Comment 1-4

Figure 3C: I suggest to include in the comparison the levels of H4K20me1 at P5, given the differences in transcript levels shown in Figure 3B.

Response 1-4

We agree with your suggestion, and we stained the hippocampal sections at P5, 12w, and 24w with an antibody against H4K20me1 to check its expression levels. Consistent with the *Setd8* mRNA expression levels (Fig. 3B), we observed a decline in H4K20me1 modification levels in NSCs during the aging process from P5 (as shown below, Fig. S3C and S3D).

(C) Representative images of staining for EGFP (green), H4K20me1 (cyan), and Hoechst (gray) in the hippocampal DG at P5, 12w, and 24w. Arrows indicate Nestin+ NSCs. Scale bar, 20 μm.

(D) Bar graph showing H4K20me1 fluorescence intensity normalized by background in NSCs, with the fluorescence intensity at P5 set to 1 (n = 30 cells from three mice per group). * $p < 0.01$ by Wilcoxon rank-sum test.

We have amended the text to explain our obtained result in the revised manuscript as follows (page 12, line 259 – page 13, line 267):

“To investigate the level of H4K20me1 modification in NSCs with age, we stained brain sections with antibodies for H4K20me1 and Nestin, and observed a decline in H4K20me1 levels in Nestin-positive (Nestin+) NSCs from P5 (S3C, and S3D). To further analyze H4K20me1 levels in quiescent and active NSCs, we stained brain sections from Nestin-EGFP mice with antibodies for H4K20me1, Ki67, and EGFP, identifying EGFP+ NSCs with radial glia-like morphology in quiescent (Ki67–) and active (Ki67+) states (Figures 3C, 3D). H4K20me1 levels decreased with age, consistent with *Setd8* expression, regardless of NSC activity state.”

Comment 1-5

Page 14 (line 295): why is LaminB1 expression invoked here, please explain better.

Response 1-5

We appreciate your comment and have clarified the rationale for invoking LaminB1 expression in this context as follows (page 15, lines 325-332):

“LaminB1 loss disrupts nuclear envelope integrity and chromatin organization, leading to altered gene expression and activation of cellular senescence pathways.^{17,18} Prolonged (6.5 months) loss of LaminB1 has been reported to cause premature aging of mouse hippocampal NSCs in middle age.¹⁸ These findings prompted us to investigate whether Setd8 might induce premature NSC aging when its expression is reduced over an extended period, starting at an early life stage (i.e., young adult stage). To address this, we utilized Setd8 cKO mice and heterozygous *Setd8* conditional knockout (het-cKO; Setd8^{flox/wt}) mice carrying Nestin-CreERT2 and Rosa26-YFP reporter transgenes.“

Comment 1-6

Figure 5G: can the authors better clarify whether genes that are deregulated upon acute Setd8 KD and overlap with genes that change in NSCs with aging, change expression in the same direction? This is an important point if Setd8 mediates aging-related changes.

Response 1-6

In response to this comment, we re-analyzed the dataset and found that genes down-regulated by *Setd8* KD significantly overlap with genes down-regulated in NSCs with aging, but not with genes up-regulated. On the other hand, genes up-regulated by *Setd8* KD significantly overlapped with genes up-regulated in NSCs with aging, and to a lesser extent with genes down-regulated (Fig. 5G, as follows). These results suggest that *Setd8* KD induces gene expression changes that resemble those observed in aging NSCs, indicating that at least some age-dependent changes in gene expression in hippocampal NSCs are mediated by decreased *Setd8* expression.

We have added this new figure and also revised the figure legend and the main text (page 17, lines 370–378) as follows:

Figure legend:

Enrichment analysis of up- (red) or downregulated (blue) DEGs of NSCs with age *in vivo*, identified in Figure 1E, on *Setd8*-KD-induced up- or downregulated genes. The leftmost bar represents the expected overlap based on a random gene set, while the remaining bars show the fold enrichment of actual overlaps.

Main text:

“We also checked the enrichment of *Setd8*-KD-induced DEGs in age-related gene expression changes in NSCs (Figure 1E) compared to a random gene set (Figure 5G). We found that genes downregulated by *Setd8* KD significantly overlap with genes downregulated in NSCs with aging, but not with genes upregulated. On the other hand, genes upregulated by *Setd8* KD significantly overlapped with genes upregulated in NSCs with aging, and to a lesser extent with genes downregulated. These results suggest that *Setd8* KD induces gene expression changes that resemble those observed in aging NSCs, indicating that at least some age-dependent changes in gene expression in hippocampal NSCs are mediated by decreased *Setd8* expression.”

Comment 1-7

Figure 5 (I-N and P): graphs show regulation of individual genes from the genomics data. These should be individually validated by qPCR using a better model and approach (related to my main point above).

Response 1-7

In response to this comment, we investigated the effects of *Setd8* knockdown using a lentiviral *Setd8* knockdown vector, as well as *Setd8* deletion using Cas9, on the proliferation of adult hippocampal NSCs. For more details, please refer to Response 1-1.

Comment 1-8

Figure 5 (O-P): can the authors provide a better description of the experiment in figure legend (long-term suppression of Setd8 at 7 days post-infection).

Response 1-8

In accordance with your suggestion, we have added explanatory details as follows (page 61, lines 1428–1436):

“(O) GSEA results showing significant enrichment of the senescence-associated gene set, as reported by Saul *et al.*, 2023, in Ctrl (white) and KD (gray) NSCs after long-term suppression of *Setd8*, observed 7 days post-infection.

(P) Expression level of interferon signaling-related genes in Ctrl (white) and KD (gray) NSCs after long-term suppression of *Setd8*, observed 7 days post-infection. * $p < 0.01$.”

Comment 1-9

Figure 2F: typo "Gained-pen"

Response 1-9

Thank you for pointing out this typo. The term "Gained-pen" in Figure 2F has been corrected to "Gained-open" in the revised manuscript.

Laboratory of Neural Regeneration and Brain Repair

Division of Biological Science
Graduate School of Science and Technology,
Nara Institute of Science and Technology (NAIST)
8916-5, Takayama-cho, Ikoma, Nara, 630-0192, Japan
Tel +81-743-72-5556

To THE *EMBO JOURNAL* -11-

Comment 2-0

This is interesting because this kind of cell is involved in sustaining adult neurogenesis and, downstream from this, the processing of memory.

Response 2-0

We thank you for this positive comment. Before conducting the revision experiments, we engaged in careful discussions with the editor to ensure that our approach to these experiments was appropriate. We hope that our additional experiments have successfully addressed the points you raised.

Major concerns:

Comment 2-1

*Can the authors clarify by which event(s) the NSCs are less in the *Setd8* mutant? Data seem ruled out differentiation. Is it for Caspase-independent cell death? Is it for stop the doublings (the time window seems too short for this but approaches like the BrdU labelling during the TAM administration may help).*

Response 2-1

To address this comment, we investigated the impact of *Setd8* depletion on NSC differentiation by employing lineage-tracing of YFP+ NSCs. At 6 weeks after tamoxifen treatment, we found that the proportion of YFP+ NeuN+ mature neurons and YFP+ Dcx+ immature neurons was significantly reduced in *Setd8* cKO mice compared to Ctrl mice (new Figures S5G, S5K, and S5L), along with a reduction in YFP+ neuronal projections to the CA3 region (new Figure S5J). Additionally, a slight decrease in astrocyte differentiation was observed (new Figures S5H, S5I, S5M, and S5N), and a larger fraction of NSCs remained undifferentiated (new Figures S5H and S5O), suggesting that *Setd8* deletion impairs NSC differentiation.

Since both NSC proliferation and neuronal differentiation were impaired, these events likely contributed to the reduction in newly generated neurons. However, the rapid and prominent loss of NSCs within one day of tamoxifen administration suggests that

impaired proliferation alone cannot fully explain NSC depletion.

New Figures S5D and S5G–O are as follows:

Figure legend:

(D) Experimental scheme for assessing the effect of *Setd8* expression decrease on NSC differentiation in the DG at 14w.

(G) Representative images of staining for YFP (green), NeuN (red), and Hoechst (gray; insets) of the DGs in Ctrl (n = 4 animals) and *Setd8* cKO (n = 4 animals). Arrowheads indicate YFP-labeled mature neurons generated from NSCs. Bottom panels show higher magnification of dashed boxes in top panels. Scale bar, 100 μ m.

Laboratory of Neural Regeneration and Brain Repair

Division of Biological Science

Graduate School of Science and Technology,
Nara Institute of Science and Technology (NAIST)
8916-5, Takayama-cho, Ikoma, Nara, 630-0192, Japan
Tel +81-743-72-5556

To THE *EMBO JOURNAL* -13-

(H and I) Representative images of staining for YFP (green), S100 β (red), GFAP (cyan), and Hoechst (gray) of the DGs in Ctrl (n = 4 animals) and *Setd8* cKO (n = 4 animals). White and yellow arrowheads indicate YFP+ GFAP+ S100 β - NSCs in the SGZ (H) and YFP+ GFAP+ S100 β - astrocytes in the granular cell layer (GCL) (H), respectively. White arrowheads indicate YFP+ GFAP+ S100 β + astrocytes in the molecular layer (ML) (I). Scale bar, 50 μ m.

(J) Representative images of staining for projection of YFP+ newly generated neurons into the CA3 region. White arrowheads highlight YFP+ cell fiber. Scale bar, 100 μ m.

(K-O) Proportion of YFP+ NeuN+ mature neurons (K), YFP+ DCX+ immature neurons (L), and YFP+ GFAP+ S100 β - astrocytes in the GCL (M), YFP+ GFAP+ S100 β + astrocytes in the ML (N), or YFP+ GFAP+ S100 β - NSCs (O) among YFP+ cells in the DG (n = 4 animals per group). **p* < 0.05 by Student's t test.

In view of these considerations, we have modified the main text as follows (page 14, line 303 – page 15, line 319):

“To further investigate the impact of *Setd8* depletion on NSC differentiation, we examined differentiation trends of YFP+ NSCs and found that the proportion of YFP+ NeuN+ mature neurons and YFP+ Dcx+ immature neurons was significantly reduced in *Setd8* cKO mice compared to Ctrl mice (Figures S5G, S5K and S5L), indicating that neuronal differentiation was impaired in the absence of *Setd8*. Supporting this observation, *Setd8* cKO mice also exhibited a reduction in projections of YFP-positive neurons to the CA3 region (Figures S5J). Additionally, a slight decrease in the proportion of astrocyte-differentiated cells was observed in *Setd8* cKO mice (Figures S5H, S5I, S5M and S5N). Notably, a larger fraction of NSCs remained undifferentiated in *Setd8* cKO mice (S5H and S5O), suggesting that *Setd8* deletion impairs the transition from NSCs to differentiated neuronal and glial lineages. Since both NSC proliferation and neuronal differentiation were impaired, these factors likely contributed to the reduction in newly

generated neurons. However, the rapid and prominent decrease in total NSC number observed one day after tamoxifen administration suggests that NSC loss cannot be fully explained by impaired proliferation alone. Therefore, we cannot exclude the possibility that caspase-independent cell death also contributes to the depletion of NSCs.”

Comment 2-2

Other experimental settings for assessing hippocampal-related memory should be conducted. Males and female should be grouped separately.

Response 2-2

We conducted the hippocampal memory-related test with males because the behavior of female mice is affected by the estrous cycle. We have explained clearly in the Methods section of the revised manuscript that we used male mice to exclude the effects of the estrous cycle in this study and focus on the effects of impaired NSC function (page 34, lines 785–788). To further respond to your suggestion, we have conducted another hippocampal-related memory test, namely the contextual fear conditioning test. *Setd8* cKO and het-cKO mice displayed significantly decreased contextual fear memory compared with Ctrl, indicating hippocampus-dependent memory impairment (as shown below, Figures S6H and I):

(H) Experimental scheme for contextual fear conditioning test at young adult age (10w).

(I) Quantification of freezing rate in the conditioning (left graph) or testing (right graph)

Laboratory of Neural Regeneration and Brain Repair

Division of Biological Science
Graduate School of Science and Technology,
Nara Institute of Science and Technology (NAIST)
8916-5, Takayama-cho, Ikoma, Nara, 630-0192, Japan
Tel +81-743-72-5556

To THE *EMBO JOURNAL* -15-

phase. * $p < 0.05$ by ANOVA with Tukey post-hoc tests. n.s., not significant.

We have added a paragraph in the main text of our revised manuscript as follows (page 16, line 351–line 353):

“We also conducted other hippocampal-related memory test, contextual fear conditioning test. *Setd8* cKO and het-cKO mice displayed significantly decreased contextual fear memory compared with Ctrl (Figures S6H and S6I).”

We have also added a paragraph in the Methods of our revised manuscript (page 35, lines 802–812).

Comment 2-3

Genomic localization of Setd8 should be mapped and related with both H4K20me1 and ATAC.

Response 2-3

To address this comment, we attempted to validate commercially available Setd8 antibodies using brain sections from *Setd8* conditional knockout (cKO) mice. We tested three different antibodies (Abcam ab3798, Abcam ab111691, and Proteintech 14063-1-AP); however, none of them produced specific or reliable signals under our experimental conditions. In contrast, we obtained a reliable signal for H4K20me1 using an H4K20me1-specific antibody, confirming the effectiveness of our immunostaining approach. These results suggest that the available Setd8 antibodies are not suitable for our system. Furthermore, given their lack of specificity and reliability, we determined that these antibodies are not suitable for ChIP experiments.

To the best of our knowledge, while ChIP-seq studies targeting H4K20me1 have been conducted, reports on Setd8 ChIP-seq appear to be limited. This is likely due to the limitations of currently available Setd8 antibodies. Given these challenges, we would like to leave this question for a future study.

Comment 2-4

What is the link between H4K20me1 levels in the gene body and the chromatin accessibility differences (mainly closure) which are evident upon physiological aging of NSC or *Setd8* KO/KD?

Response 2-4

Alterations in H4K20me1 levels have been reported to correlate with chromatin accessibility changes, particularly with chromatin closure (Myers *et al.*, Epigenetics Chromatin 2020; PMID: 32178723). However, whether H4K20me1 directly regulates chromatin accessibility remains unclear, likely because decreased *Setd8* expression induces widespread changes in the expression of various epigenetic regulators, which may, in turn, influence chromatin structure.

To further investigate this, we reanalyzed our RNA-seq data and found that *Setd8* suppression in cultured NSCs led to substantial and statistically significant changes in the expression of multiple epigenetic regulators, including *Tet1*, *Tet2*, and several *HDAC* family genes (as shown below, new Figure S7G, Table S4 and S5). Given the known roles of these factors in DNA methylation and histone deacetylation, it is plausible that the chromatin accessibility differences observed upon physiological aging of NSCs or *Setd8* KD are not solely due to changes in H4K20me1 levels but are also influenced by downstream epigenetic pathways regulated by *Setd8*.

Laboratory of Neural Regeneration and Brain Repair

Division of Biological Science
Graduate School of Science and Technology,
Nara Institute of Science and Technology (NAIST)
8916-5, Takayama-cho, Ikoma, Nara, 630-0192, Japan
Tel +81-743-72-5556

To THE *EMBO JOURNAL* -17-

Fig. S7G legend:

Scatter plots showing the expression levels of genes associated with epigenetic modification and chromatin remodeling (n = 171) in control and *Setd8* KD NSCs at day 3 (left) and day 7 (right). Up- (red) and downregulated (blue) differentially expressed genes are highlighted.

According to these results, we have also amended main text as follows (page 18, line 405 – page19, line 412):

“To further investigate the downstream effects of *Setd8* downregulation, we checked expression of epigenetic gene using our RNA-seq data and found that *Setd8* downregulation in NSCs led to significant changes in the expression of multiple epigenetic regulators, including *Tet1*, *Tet2*, and several *HDAC* family genes (Figure S7G, Tables S4 and S5). These results suggest that *Setd8* functions in coordination with other epigenetic regulators and that, beyond its role in H4K20me1 modification, *Setd8* contributes to broader epigenomic regulation, influencing chromatin dynamics and cell proliferation in accordance with previous reports.^{29,55}”

Comment 2-5

*An interesting question would be: are the changes caused by *Setd8* repression (linked with senescence) reversible if *Setd8* comes back to normal levels? This would imply that the NSC aging in DG may be slow down (or reverted) with *Setd8* supplementation. Can the authors address this question experimentally?*

Response 2-5

Since the long-term *Setd8* downregulation in NSCs may have caused nearly irreversible changes in chromatin modification, likely due to cellular senescence, we conducted experiments using an inhibitor that can immediately restore *Setd8* activity without altering its protein levels. Treatment with the inhibitor effectively reduced H4K20me1 levels and suppressed NSC proliferation. Importantly, upon inhibitor removal,

H4K20me1 levels returned to normal, and NSC proliferation was restored (as below, Figure S9A–F). These data suggest that the reduction in NSC proliferation caused by a transient reduction in Setd8 activity is reversible once Setd8 function is restored. Therefore, the early stages of aging, when *Setd8* expression begins to decline, may represent a critical window for intervention.

(A) Schematic representation of the experimental timeline. NSCs derived from embryonic cortex with repeated passages (B and C) and adult hippocampus (E and F) were plated

Laboratory of Neural Regeneration and Brain Repair

Division of Biological Science
Graduate School of Science and Technology,
Nara Institute of Science and Technology (NAIST)
8916-5, Takayama-cho, Ikoma, Nara, 630-0192, Japan
Tel +81-743-72-5556

To THE EMBO JOURNAL -19-

and treated with either DMSO (control) or the Setd8 inhibitor S8i (5 μ M) for 2 days, followed by maintenance in either DMSO or S8i for an additional 2 days before fixation.

- (B) Representative immunofluorescence images of NSCs stained for Nestin (green) and H4K20me1 (cyan) with Hoechst counterstaining (gray) under different treatment conditions: DMSO, S8i, and S8i withdrawal (DMSO→S8i). Scale bar: 50 μ m.
- (C) Representative immunofluorescence images showing Nestin (green) and EdU incorporation (magenta) with Hoechst counterstaining (gray) to assess proliferation of NSCs under different treatment conditions. Scale bar: 50 μ m.
- (D) Quantification of EdU⁺ cells among Nestin⁺ NSCs in panel (C). Data are presented as mean \pm SEM. Statistical significance was determined using a one-way ANOVA followed by post-hoc tests (* p < 0.05, n.s.: not significant).
- (E) Representative immunofluorescence images showing Nestin (green) and EdU incorporation (red) with Hoechst counterstaining (gray) to assess proliferation of adult hippocampal NSCs under different treatment conditions. Scale bar: 50 μ m.
- (F) Quantification of EdU⁺ cells among Nestin⁺ adult hippocampal NSCs in panel (E). Data are presented as mean \pm SEM. Statistical significance was determined using a one-way ANOVA followed by post-hoc tests (* p < 0.05).

We have also modified our main text as follows (page 21, lines 461–472):

“Finally, we investigated whether the proliferation of NSCs, which had been reduced due to Setd8 dysfunction, could be restored. Since the long-term *Setd8* downregulation in NSCs may have caused nearly irreversible changes in chromatin modification, likely due to cellular senescence, we conducted experiments using the selective Setd8 inhibitor UNC037927 that can immediately restore Setd8 activity without altering its protein levels (Figure S9A). Treatment with the inhibitor effectively reduced H4K20me1 levels and suppressed NSC proliferation. Importantly, upon inhibitor removal, H4K20me1 levels returned to normal, and NSC proliferation was restored (Figure S9B–

F). These data suggest that the reduction in NSC proliferation caused by a transient reduction in Setd8 activity is reversible once Setd8 function is restored. Therefore, the early stages of aging, when Setd8 expression begins to decline, may represent a critical window for intervention.”

Minor Points:

Comment 2-6

Since it is true that the number of NSCs at 10W in mutants is comparable of the number at 30W in WT animals I would not state that the biological aging is accelerated (Pag 14). In fact the profiling of the aging -associated molecular and phenotypic signatures is missing there.

Response 2-6

We appreciate this insightful comment and have carefully considered your suggestion. In response, we have revised the text as follows (page 16, lines 339-340) to provide a clearer and more precise description of our observations.

Previous text:

“implying that biological aging had been accelerated in the cKO mice.”

Revised text:

“indicating an accelerated loss of NSCs in cKO mice.”

Comment 2-7

Fig.5G and H: it is not entirely clear to me what is this analysis. Could you please clarify?

Response 2-7

Figure 5G illustrates the relationship between age-related gene expression

Laboratory of Neural Regeneration and Brain Repair

Division of Biological Science
Graduate School of Science and Technology,
Nara Institute of Science and Technology (NAIST)
8916-5, Takayama-cho, Ikoma, Nara, 630-0192, Japan
Tel +81-743-72-5556

To THE *EMBO JOURNAL* -21-

changes in NSCs (identified in Figure 1E) and *Setd8* KD-induced differentially expressed genes (DEGs). The leftmost bar represents the expected overlap based on a random gene set, while the remaining bars show the fold enrichment of actual overlaps between age-related DEGs and *Setd8* KD DEGs. Statistically significant enrichment ($p < 0.01$, hypergeometric test) suggests that *Setd8* KD influences gene expression patterns associated with aging. Figure 5H presents a similar analysis but focuses on genes involved in NSC quiescence or activation, rather than age-related DEGs, and their overlap with *Setd8* KD-induced DEGs. This distinction highlights how *Setd8* KD affects not only transcriptional changes associated with aging but also pathways regulating NSC states. To improve clarity, we have amended the main text and figure legends (as below) for Figures 5G and 5H to better reflect these points. We appreciate your insights and have made the necessary clarifications accordingly.

Revised main text (page 17, lines 370–378):

“We also checked the enrichment of *Setd8*-KD-induced DEGs in age-related gene expression changes in NSCs (Figure 1E) compared to a random gene set (Figure 5G). We found that genes downregulated by *Setd8* KD significantly overlap with genes downregulated in NSCs with aging, but not with genes upregulated. On the other hand, genes upregulated by *Setd8* KD significantly overlapped with genes upregulated in NSCs with aging, and to a lesser extent with genes downregulated. These results suggest that *Setd8* KD induces gene expression changes that resemble those observed in aging NSCs, indicating that at least some age-dependent changes in gene expression in hippocampal NSCs are mediated by decreased *Setd8* expression.”

Revised figure legend (page 61, lines 1428–1436):

“(G) Enrichment analysis of up- (red) or downregulated (blue) DEGs of NSCs with age *in vivo*, identified in Figure 1E, on *Setd8*-KD-induced up- or downregulated genes. The leftmost bar represents the expected overlap based on a random gene set, while the

remaining bars show the fold enrichment of actual overlaps. $*p < 0.01$ by hypergeometric distribution.

(H) Enrichment analysis of NSC quiescence- (red) or activation-associated (blue) genes, reported by Shin *et al.*, 2015, on *Setd8*-KD-induced up- or downregulated genes. The leftmost bar represents the expected overlap based on a random gene set, while the remaining bars show the fold enrichment of actual overlaps. $*p < 0.001$ by hypergeometric distribution.”

Comment 2-8

Fig.5: Describe better in the figure the timing of the experiments (improve the labelling). As it is, the figure is misleading for the reader.

Response 2-8

Following your suggestion, we have modified Fig. 5 and added explanatory details in the figure legend as follows (page 62, lines 1440-1444):

“(O) GSEA results showing significant enrichment of the senescence-associated gene set, as reported by Saul *et al.*, 2023, in Ctrl (white) and KD (gray) NSCs after long-term suppression of *Setd8*, observed 7 days post-infection.

(P) Expression level of interferon signaling-related genes in Ctrl (white) and KD (gray) NSCs after long-term suppression of *Setd8*, observed 7 days post-infection. $*p < 0.01$.”

Comment 2-9

Fig. 6D: not entirely clear what group of genes are "with H4K20me1 modification in their gene body and with RPKM values less than $< 0.5 \times$ those of Ctrl in the Setd8 KD NSCs"

Response 2-9

We sincerely appreciate your comment and apologize for the oversight in our original description. We have corrected the text as follows (page 19, lines 425–427):

Laboratory of Neural Regeneration and Brain Repair

Division of Biological Science
Graduate School of Science and Technology,
Nara Institute of Science and Technology (NAIST)
8916-5, Takayama-cho, Ikoma, Nara, 630-0192, Japan
Tel +81-743-72-5556

To THE EMBO JOURNAL -23-

"Genes located in genomic regions where H4K20me1 modification levels in the gene body decreased by more than 2-fold in *Setd8* KD NSCs significantly overlapped with downregulated DEGs in *Setd8* KD NSCs (Figure 6D)."

Comment 2-10

Please use the correct international nomenclature for genes.

Response 2-10

We appreciate your feedback and have carefully revised the manuscript to ensure that gene names adhere to the correct international nomenclature.

Additional suggestions:

Comment 2-11

*scRNA+ATAC should be done in the *Setd8* KO too.*

Response 2-11

We had hoped to do that as well, but, unfortunately, it was difficult to perform even bulk RNA-seq or ATAC-seq with adequate quality using *Setd8* cKO mice. Because the number of NSCs in *Setd8* cKO mice was insufficient, isolating NSCs in these mice was even more difficult than in the 30-week-old Nestin-EGFP mice. Given the current limitations of spatial omics analysis for simultaneous mRNA and ATAC profiling, we would like to leave this question for a future study.

Laboratory of Neural Regeneration and Brain Repair

Division of Biological Science
Graduate School of Science and Technology,
Nara Institute of Science and Technology (NAIST)
8916-5, Takayama-cho, Ikoma, Nara, 630-0192, Japan
Tel +81-743-72-5556

To THE *EMBO JOURNAL* -24-

We hope that we have addressed the reviewers' comments comprehensively and satisfactorily, and await your final decision regarding the revised manuscript.

Sincerely yours,

Taito Matsuda

Taito Matsuda, Ph.D.

Associate Professor

Laboratory of Neural Regeneration and Brain Repair,
Division of Biological Science, Graduate School of
Science and Technology,
Nara Institute of Science and Technology (NAIST),
8916-5, Takayama-cho, Ikoma, Nara, 630-0192, Japan

Tel +81-743-72-5556

e-mail matsuda.taito@naist.ac.jp

Dear Dr Matsuda,

Thank you for submitting your revised manuscript (EMBOJ-2024-118177R) to The EMBO Journal. Your amended study was sent back to the referees for their scientific re-evaluation, and we have received detailed comments from both of them, which I enclose below. As you will see, the experts state that the work has been substantially enhanced by the revisions and they are now broadly in favour of publication, pending minor revision.

Thus, we are pleased to inform you that your manuscript has been accepted in principle for publication in The EMBO Journal.

We now need you to take care of a number of issues related to formatting and data presentation as detailed below, which should be addressed at re-submission.

Please contact me at any time if you have additional questions related to below points.

As you might have noted from our webpage, every paper at the EMBO Journal now includes a 'Synopsis', displayed on the html and freely accessible to all readers. The synopsis includes a 'model' figure as well as 2-5 one-short-sentence bullet points that summarize the article. I would appreciate if you could provide this figure and the bullet points.

Thank you for giving us the chance to consider your manuscript for The EMBO Journal. I look forward to your final revision.

Again, please contact me at any time if you need any help or have further questions.

Best regards,

Daniel Klimmeck

>> Author Contributions: Remove the author contributions information from the manuscript text. Note that CRediT has replaced the traditional author contributions section as of now because it offers a systematic machine-readable author contributions format that allows for more effective research assessment. and use the free text boxes beneath each contributing author's name to add specific details on the author's contribution.

More information is available in our guide to authors.
<https://www.embopress.org/page/journal/14602075/authorguide>

>> Figure callouts: Please ensure that the figure panel Fig 6H is called out in sequential order.

>> Reagents and Tools table: please provide a Reagents and Tools table as a separate file using the existing template in the Guide For Authors, listing key reagents, experimental models, software and relevant equipment.

>> Section order should be as follows: title page with complete author information, abstract, keywords, introduction, results, discussion, methods, data availability section, acknowledgements, disclosure and competing interests statement, references, main figure legends, tables, expanded figure legends.´

>> References: adjust reference format to EMBO Journal format, 10 authors et al, and place References after the Discussion, before figure legends.

>> Funding: please enter the following funding information to our online system: 'Nakajima Foundation and the Mochida Foundation as well as the JST971 FOREST Program, Grant Number JPMJFR231Z'.

>> Data availability section: remove the referee token and make sure GEO dataset are made publicly accessible. Add a URL link to the dataset.

>> Avoid textual redundancy in the introduction, results and discussion sections with your earlier 2021 study (PMID 34526402).

>>Appendix file with ToC: the file with suppl. information should be renamed "Appendix" and uploaded as a PDF. Please rename the suppl. tables "Appendix Table S1" etc., and the suppl. figures "Appendix Figure S1", and adjust callouts in the main text and figure legends accordingly. Please add a table of contents, including page numbers.

>> Author checklist: please annotate the primary GEO datasets in the 'Data Availability' section.

>> Source data: as to our journal policies, we kindly request you to

>>>> enhance data resolution in the entire appendix file.

>>>> provide uncropped micrograph source data for Figures S1, S3, S4, S5, S6 and S8.

>> Consider additional changes and comments from our production team as indicated below:

- DAS:

Please note that the specific URLs for GSE256417 dataset is not provided in the data availability statement.

- Figure legends:

1. Please indicate what */ **/ ***/ **** represents; if this represents p value(s), please indicate the statistical test used and where appropriate, specify the exact p value in the legend(s) of figure(s) 6G.

2. Please note that the exact p values are not provided in the legends of figures 1D, 2C, 3B, D, G, H, I, J; 4E, F, G, I; 5G, H, P; 6C, H, I.

3. Please indicate the statistical test used for data analysis in the legends of figures 3B, 5B-D, G, H, P; 6F.

4. Please note that the box plots need to be defined in terms of minima, maxima, centre, bounds of box and whiskers, and percentile in the legends of figures 1D, 2C, 3D, 6C, H, I

5. Please note that information related to n is missing in the legends of figures 1D, 4I, 5I-N, P; 6H, I.

6. Please note that the error bars are not defined in the legends of figures 4E, F, G, I; 5I-N, P

Referee #1:

I have now carefully gone through the response to my initial comments to the manuscript, and found that they were all addressed properly by the authors, either experimentally and/or in writing.

Referee #2:

I am satisfied with the authors' response to the points I raised. Almost all of my suggestions have been addressed, except for those affected by technical limitations. The manuscript is now more robust and complete, and I therefore recommend its acceptance.

The authors addressed the remaining editorial issues.

Dear Dr Matsuda,

Thank you for submitting the revised version of your manuscript. I have now evaluated your amended manuscript and concluded that the remaining minor concerns have been sufficiently addressed.

I am thus pleased to inform you that your manuscript has been accepted for publication in the EMBO Journal.

On a different note, I would like to alert you that EMBO Press offers a format for a video-synopsis of work published with us, which essentially is a short, author-generated film explaining the core findings in hand drawings, and, as we believe, can be very useful to increase visibility of the work. Please see the following link for representative examples and their integration into the article web page:

<https://www.embopress.org/doi/full/10.15252/emj.2019103932>

Best regards,

Daniel Klimmeck

Daniel Klimmeck, PhD
Senior Editor
The EMBO Journal
EMBO
Postfach 1022-40
Meyerohofstrasse 1
D-69117 Heidelberg
contact@embojournal.org